# Integrin-based diffusion barrier separates membrane domains enabling the formation of microbiostatic frustrated phagosomes

Michelle E Maxson[1], Xenia Naj[2], Teresa R O'Meara[3], Jonathan D Plumb[1], Leah E Cowen[3], Sergio Grinstein[1,4,5]*

[1]Program in Cell Biology, Hospital for Sick Children, Toronto, Canada; [2]Institute for Medical Microbiology, Virology and Hygiene, University Medical Center Hamburg-Eppendorf, Hamburg, Germany; [3]Department of Molecular Genetics, University of Toronto, Toronto, Canada; [4]Keenan Research Centre for Biomedical Science, St. Michael's Hospital, Toronto, Canada; [5]Department of Biochemistry, University of Toronto, Toronto, Canada

**Abstract** *Candida albicans* hyphae can reach enormous lengths, precluding their internalization by phagocytes. Nevertheless, macrophages engulf a portion of the hypha, generating incompletely sealed tubular phagosomes. These frustrated phagosomes are stabilized by a thick cuff of F-actin that polymerizes in response to non-canonical activation of integrins by fungal glycan. Despite their continuity, the surface and invaginating phagosomal membranes retain a strikingly distinct lipid composition. PtdIns(4,5)$P_2$ is present at the plasmalemma but is not detectable in the phagosomal membrane, while PtdIns(3)P and PtdIns(3,4,5)$P_3$ co-exist in the phagosomes yet are absent from the surface membrane. Moreover, endo-lysosomal proteins are present only in the phagosomal membrane. Fluorescence recovery after photobleaching revealed the presence of a diffusion barrier that maintains the identity of the open tubular phagosome separate from the plasmalemma. Formation of this barrier depends on Syk, Pyk2/Fak and formin-dependent actin assembly. Antimicrobial mechanisms can thereby be deployed, limiting the growth of the hyphae.
DOI: https://doi.org/10.7554/eLife.34798.001

**\*For correspondence:**
sergio.grinstein@sickkids.ca

**Competing interests:** The authors declare that no competing interests exist.

## Introduction

*Candida albicans* is a commensal fungus that colonizes the epithelial surfaces of 30–70% of healthy individuals (*Perlroth et al., 2007*). However, in immune-compromised individuals, *C. albicans* can cause invasive, life-threatening disease. The mortality rate for infected patients is 46–75%, with candidiasis classified as the fourth most common nosocomial bloodstream infection (*Brown et al., 2012*). Invasive candidiasis is correlated with a switch of *C. albicans* from its yeast form to a hyphal form, a shift that can be induced in vitro by nutrient deprivation among other cues (reviewed in *Sudbery, 2011*). In vivo, *C. albicans* hyphae are capable of invading epithelium and endothelium; in addition *C. albicans* is capable of forming recalcitrant biofilms and inducing inflammation (*Sudbery, 2011*). These conditions activate host defense mechanisms for the control and clearance of *C. albicans*, mounted predominantly by phagocytic cells of the innate immune system.

Phagocytes can effectively sense, internalize and kill invasive *C. albicans*. Accordingly, impairment of the phagocytic response, e.g. by elimination of macrophages and neutrophils, is associated with disseminated candidiasis (reviewed in *Netea et al., 2015*). Phagocytic cells possess receptors that bind the *C. albicans* cell wall and trigger uptake of the fungus into a phagosome. The *C. albicans*

**eLife digest** Billions of microorganisms live on, and in, the human body. Known as the human microbiome, most of these microscopic hitchhikers are harmless. But, for people with a compromised immune system, common species can sometimes cause disease. For example, the yeast *Candida albicans*, which colonises between 30 and 70% of the population, is normally harmless, but can switch to a disease-causing version that makes branching structures called hyphae. These hyphae grow fast, piercing and damaging the tissues around them.

Immune cells called macrophages usually engulf invading microbes. These cells recognise sugars on the outside of *C. albicans*, and respond by wrapping their membranes around the yeast, drawing the microorganism in, and sealing it into closed structures called phagosomes. Then, the macrophages fill the phagosomes with acid, enzymes and destructive chemicals, which breaks the yeast down. Yet, *C. albicans* hyphae grow larger than macrophages, making them difficult to control.

Maxson et al. have now tracked the immune response revealing how macrophages try to control large hyphae. The immune cells were quick to engulf *C. albicans* in its normal yeast form, but the response slowed down in the presence of hyphae. Electron microscopy revealed that the large structures were only partly taken in. Rather than form a closed phagosome, the macrophages made a cuff around the middle of the hypha, leaving the rest hanging out.

The process starts with a receptor called CR3, which detects sugars on the outside of the hyphae. CR3 is a type of integrin, a molecule that sends signals from the surface to the inside of the immune cell. A network of filaments called actin assemble around the hypha, squeezing the membrane tight. The macrophage then deploys free radicals and other damaging chemicals inside the closed space. The seal is not perfect, and some molecules do leak out, but the effect slows the growth of the yeast. When a phagosome cannot engulf an invading microbe, a state that is referred to as being "frustrated", the leaking of damaging chemicals can harm healthy tissues and lead to inflammation and disease.

These findings reveal that macrophages do at least try to form a complete seal before releasing their cocktail of chemicals. Understanding how the immune system handles this situation could open the way for new treatments for *C. albicans* infections, and possibly similar diseases related to "frustrated engulfment" (such as asbestos exposure, where asbestos fibers are also too large to engulf). However, one next step will be to find out what happens to partly engulfed hyphae, and how this differs from the fate of fully engulfed yeast.

DOI: https://doi.org/10.7554/eLife.34798.002

cell wall is composed mostly (80–90%) of polysaccharides, containing $\approx$ 60% β-(1,3) and -(1,6) glucans, and $\approx$ 40% O- and N-linked mannans (*Ruiz-Herrera et al., 2006*). As such, the main non-opsonic phagocytic receptors for *C. albicans* are the C-type lectin family of receptors, including Dectin1, the mannose receptor, and DC-SIGN (reviewed in *Hardison and Brown, 2012*). The phagosome typically matures rapidly after closure, evolving into an acidic, degradative and microbicidal compartment. Acquisition of antimicrobial properties by this compartment depends on its ability to accumulate and retain toxic compounds, including reactive oxygen species (ROS). Superoxide produced by the NADPH oxidase undergoes dismutation into hydrogen peroxide in the acidic luminal environment generated by the V-ATPase, which additionally favors the catalytic activity of various hydrolases. Transporters such as NRAMP-1, that antagonize microbial growth by depleting the phagosome of nutrients, also depend on phagosomal $H^+$ for the extrusion of metal ions.

Unlike most other microbes, *C. albicans* presents a distinct problem for phagocytes. The hyphal form of *C. albicans* can grow at a rate of 18.8 μm hr$^{-1}$ (*GOW and Gooday, 1982*), quickly exceeding the size of the phagocytes themselves. The challenge is greatest for macrophages, which migrate to infection sites later than the polymorphonuclear cells, and thus encounter growing hyphae (reviewed in *Erwig and Gow, 2016*). Despite being remarkably plastic, macrophages have difficulty engulfing the much larger *C. albicans* hyphae, an impasse that no doubt contributes to the pathogenesis of candidiasis.

The aim of the current study was to examine the dynamic and complex process of *C. albicans* phagocytosis by macrophages. We found that attempts to engulf large hyphae result in the formation of incomplete (frustrated) phagosomes, which nevertheless segregate a section of the hypha, preferentially exposing it to microbiostatic products. The mechanism and fungal components underlying the formation of the diffusion barrier established by the phagocyte when generating the frustrated phagosome was analyzed using a combination of imaging, pharmacological and genetic approaches.

## Results

### Phagocytosis of *C. albicans* hyphae

To optimize the phagocytosis of *C. albicans*, which has a cell wall rich in β-glucans (*Gow et al., 2011*), we used RAW 264.7 macrophages stably expressing the Dectin1 receptor (RAW-Dectin1; *Esteban et al., 2011*). Yeast or hyphal forms of *C. albicans* expressing BFP (*Candida*-BFP; *Strijbis et al., 2013*) were used as targets to facilitate their visualization. Under the conditions used to generate them, *C. albicans* hyphae were considerably longer (>15 µm) than the macrophages (8–10 µm in diameter). After 1 hr of co-incubation with the macrophages the yeast form was fully engulfed (*Figure 1A*), while a significant number of hyphal *C. albicans* were only partially internalized (68.5% ± 4.5, while 31.5% ± 4.6 were fully internalized; 1019 events from 12 independent experiments), which was verified using fluorescent concanavalin A to label exposed hyphae (*Figure 1B*). This was similar to the frustrated engulfment of >20 µm *C. albicans* hyphae reported earlier (*Lewis et al., 2012*). Transmission electron microscopy confirmed that most hyphae were only partially internalized (*Figure 1C*) and, in addition, revealed the existence around the neck of the frustrated phagosome of a low-contrast structure seemingly devoid of membrane-bound organelles (*Figure 1C*, inset), previously interpreted by *Strijbis et al., 2013* as accumulated actin. Indeed, this region corresponded to an actin-rich cuff-like structure (*Figure 1D*); F-actin was so highly accumulated at the cuff that the remainder of the cellular actin could only be visualized when images were overexposed (*Figure 1D*, inset). Note that the remainder (i.e. the base) of the frustrated phagocytic cup was virtually devoid of F-actin. 3D visualization verified the continuous accumulation of F-actin around the neck of the tubular phagosomes lining individual hyphae and its sharp delineation of the intracellular and extracellular portions of the fungus (*Figure 1E,F,G,H* and *Video 1*). This actin cuff was observed for RAW-Dectin1 cells engulfing *C. albicans* hyphae up to 100 µm in size (data not shown), and occurred in 96.3% ± 1.9 of the partially internalized hyphae (674 events analyzed in 12 independent experiments). These data support published accounts of actin cuff-like structures seen during the phagocytosis of various filamentous targets (*García-Rodas et al., 2011*; *Gerisch et al., 2009*; *Heinsbroek et al., 2009*; *Prashar et al., 2013*; *Strijbis et al., 2013*). The occurrence of frustrated phagocytosis with formation of a pronounced actin cuff was not unique to the RAW-Dectin1 cell line; similar features were seen when murine or human primary macrophages were confronted with *C. albicans* hyphae (*Figure 1—figure supplement 1A and B*, respectively). The actin cuff was remarkably stable, lasting for at least 90 min without contracting (*Figure 1I*). Nevertheless, the actin composing these structures undergoes measurable turnover (treadmilling), since the cuffs underwent gradual disassembly when the cells were treated with latrunculin A, which scavenges actin monomers (last two panels, *Figure 1I*). These long-lasting yet dynamic cuffs identify the frustrated phagocytic cups generated by macrophages attempting to eliminate *C. albicans* hyphae.

### Dectin1 and cadherins do not localize to the actin cuff

We proceeded to probe the receptors whose signaling could potentiate the formation of the actin cuff. Because C-type lectin signaling contributes importantly to *C. albicans* phagocytosis (*de Turris et al., 2015*; *Tafesse et al., 2015*; *Xu et al., 2009*), we analyzed whether Dectin1 accumulated in the membrane at sites where cuffs were evident. Remarkably, while Dectin1 was clearly concentrated in patches elsewhere along the frustrated phagocytic cup, it was poorly detectable by immunostaining near the actin cuff (ratio cuff: cup 0.60 ± 0.04; n = 30 p<0.0001; *Figure 2A* and inset). The failure to detect accumulation of Dectin1 at these sites was not attributable to masking of the exofacial epitope, possibly resulting from tight apposition to the hyphae, because similar results were obtained

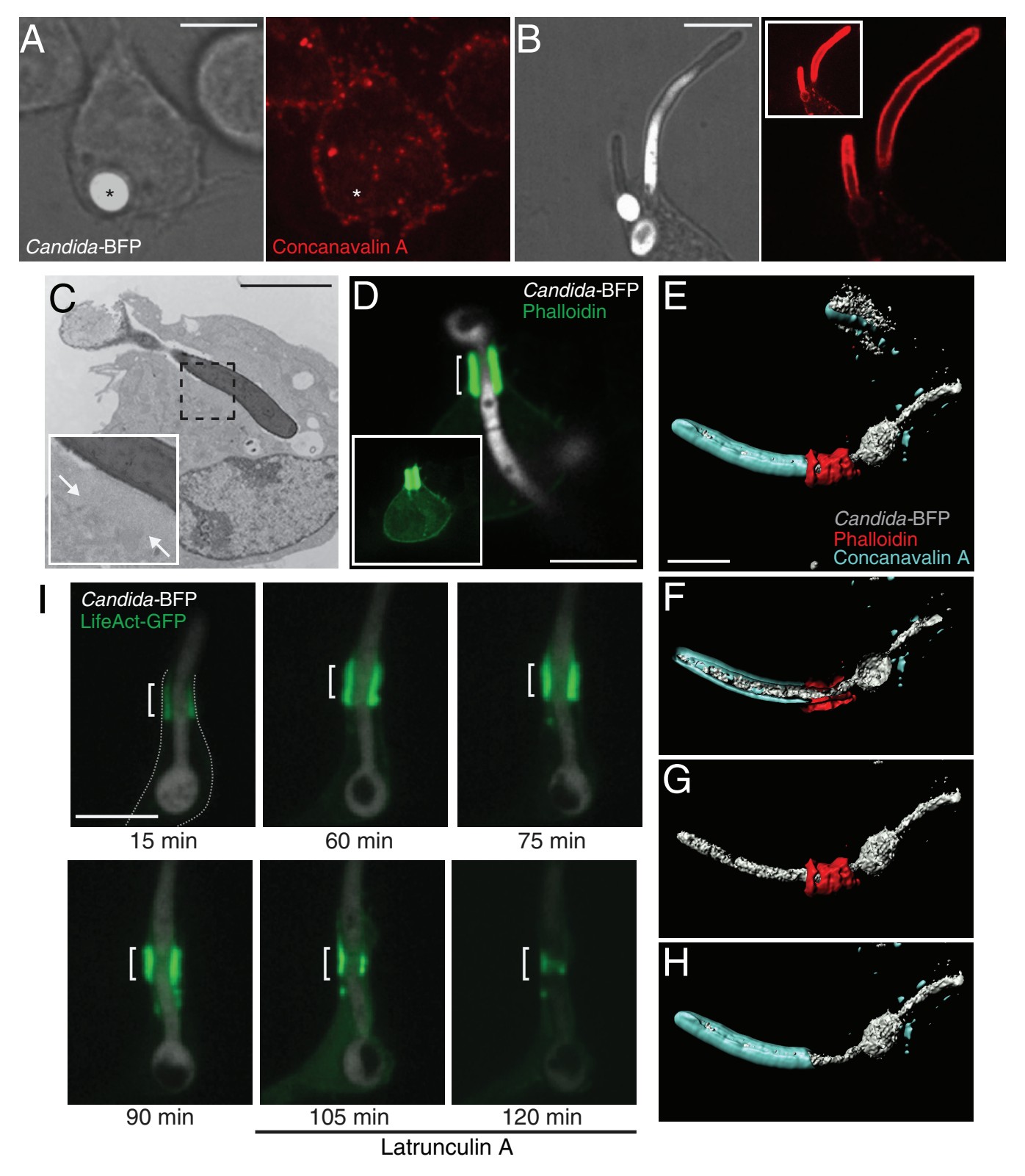

**Figure 1.** Partial phagocytosis of *C. albicans* hyphae is associated with formation of an actin cuff. Phagocytosis of *C. albicans* yeast (A) or hypha (B) by RAW-Dectin1 cells. After incubation with *Candida*-BFP, RAW-Dectin1 cells were fixed and extracellular *C. albicans* stained using Alexa594-conjugated concanavalin A (red). The fluorescence of the BFP is shown in white here and elsewhere to reveal the location of the *Candida*-BFP. Inset in (B): overexposure of the concanavalin A signal to show less intense, staining of the macrophage membrane (as in A). Scale bars: 5 μm and 10 μm,

*Figure 1 continued on next page*

*Figure 1 continued*

respectively. (C) Transmission electron micrograph of a RAW-Dectin1 cell with a partially internalized *C. albicans* hypha. Area of organelle clearance corresponding to the cuff structure is indicated in inset by arrows. Scale bar: 5 µm. (D) F-actin enrichment at the neck of partial phagosome. RAW-Dectin1 cells were allowed to internalize *C. albicans* hyphae, fixed and stained with fluorescent phalloidin (green). Actin cuff indicated with a bracket. Inset: overexposure to show the less intense cellular actin. Scale bar: 10 µm. (E–H) 3D rendering of a *C. albicans* hypha partially internalized by a RAW-Dectin1 cell. After incubation with *Candida*-BFP (white), RAW-Dectin1 cells were fixed and extracellular portions of the hyphae stained using Alexa647-conjugated concanavalin A (blue). Actin was stained with fluorescent phalloidin (red). Scale bar: 5 µm. (F) 3D rendering sliced near the middle of the tubular phagosome, (G) same as E showing only the hypha (white) and actin (red), and (H) same as E showing only the hypha (white) and concanavalin A (blue). (I) Stability of the actin cuff assessed by live cell imaging. RAW-Dectin1 cells expressing LifeAct-GFP were allowed to internalize *C. albicans* hyphae and imaged at defined intervals. Where indicated (105 min) 1 µM latrunculin A was added and recording continued. Actin cuff location indicated by bracket. Scale bar: 10 µm. Images are representative of ≥30 fields from ≥3 separate experiments of each type. In this and subsequent figures the outline of the phagocyte (when not readily apparent) is indicated by a dotted grey line.

DOI: https://doi.org/10.7554/eLife.34798.003

The following figure supplement is available for figure 1:

**Figure supplement 1.** Actin cuffs are observed in both murine and human primary macrophages infected with *C. albicans* hyphae.

DOI: https://doi.org/10.7554/eLife.34798.004

when the receptors were tagged with emerald fluorescent protein and visualized directly in live cells (ratio cuff: cup $0.56 \pm 0.04$; n = 15, $p<0.0001$; *Figure 2B* and inset).

In epithelial and endothelial cells, host E- or N-cadherin, respectively, have been reported to contribute to *C. albicans* internalization (*Moreno-Ruiz et al., 2009*). This process involved the recruitment of α- and β-catenins and activation of the Arp2/3 pathway for actin nucleation. In agreement with these reports, we observed E-cadherin and β-catenin accumulation at sites of where *C. albicans* hyphae were being internalized by epithelial A431 cells, with particular accumulation at sites where actin polymerized (*Figure 2—figure supplement 1*). We considered whether a similar mechanism was responsible for the formation of actin cuffs by macrophages. However, neither E-cadherin nor β-catenin was detectable in RAW-Dectin1 cells or in primary human macrophages by immunoblotting (*Figure 2C*) or by immunofluorescence (not illustrated). Under comparable conditions, robust signals were obtained when probing A431 cells (*Figure 2C*). When expressed heterologously in macrophages E-cadherin-GFP was found to line the surface membrane, but was absent from the phagocytic cup (*Figure 2D*), while β-catenin-GFP was largely soluble and did not accumulate at the cuff (*Figure 2E*). Thus, E-cadherin and β-catenin are unlikely to mediate phagocytosis of *C. albicans* in macrophages. Nevertheless, low levels of expression of these proteins (below the level of detection of our assays) or other cadherins may have mediated the internalization. This possibility was assessed by treating the cells with EDTA, which chelates the $Ca^{2+}$ known to be required for ligand binding by cadherins (reviewed in *Brasch et al., 2012*). As shown in *Figure 2F*, omission of $Ca^{2+}$ had no effect on actin cuff formation in *C. albicans*-infected RAW-Dectin1 cells.

## Integrin $\alpha_M \beta_2$ is involved in the formation of the actin cuff

Actin can also be tethered to the phagocytic cup via integrins (*Freeman et al., 2016*). Integrins can be directly or indirectly involved in the phagocytosis of opsonized particles, apoptotic cells and a variety of other targets (reviewed in *Dupuy and Caron, 2008*) and link with actin filaments via talin and vinculin (reviewed in *Shattil et al., 2010*). However, canonical integrin activation and ligand binding require divalent cations (reviewed in *Leitinger et al., 2000*), and would therefore be inhibited by their chelation with EDTA. Moreover, actin cuffs formed normally in CALDAG-GEF1$^{-/-}$ macrophages

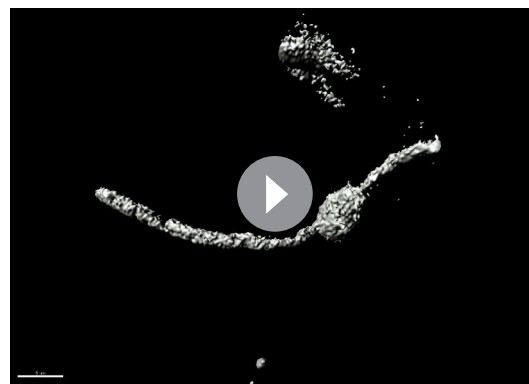

**Video 1.** 3D rendering of a RAW-Dectin1 cell with a partially internalized *Candida*-BFP hypha (white), showing the demarcation of concanavalinA (blue) by the actin cuff (red). See *Figure 1* for additional information.

DOI: https://doi.org/10.7554/eLife.34798.005

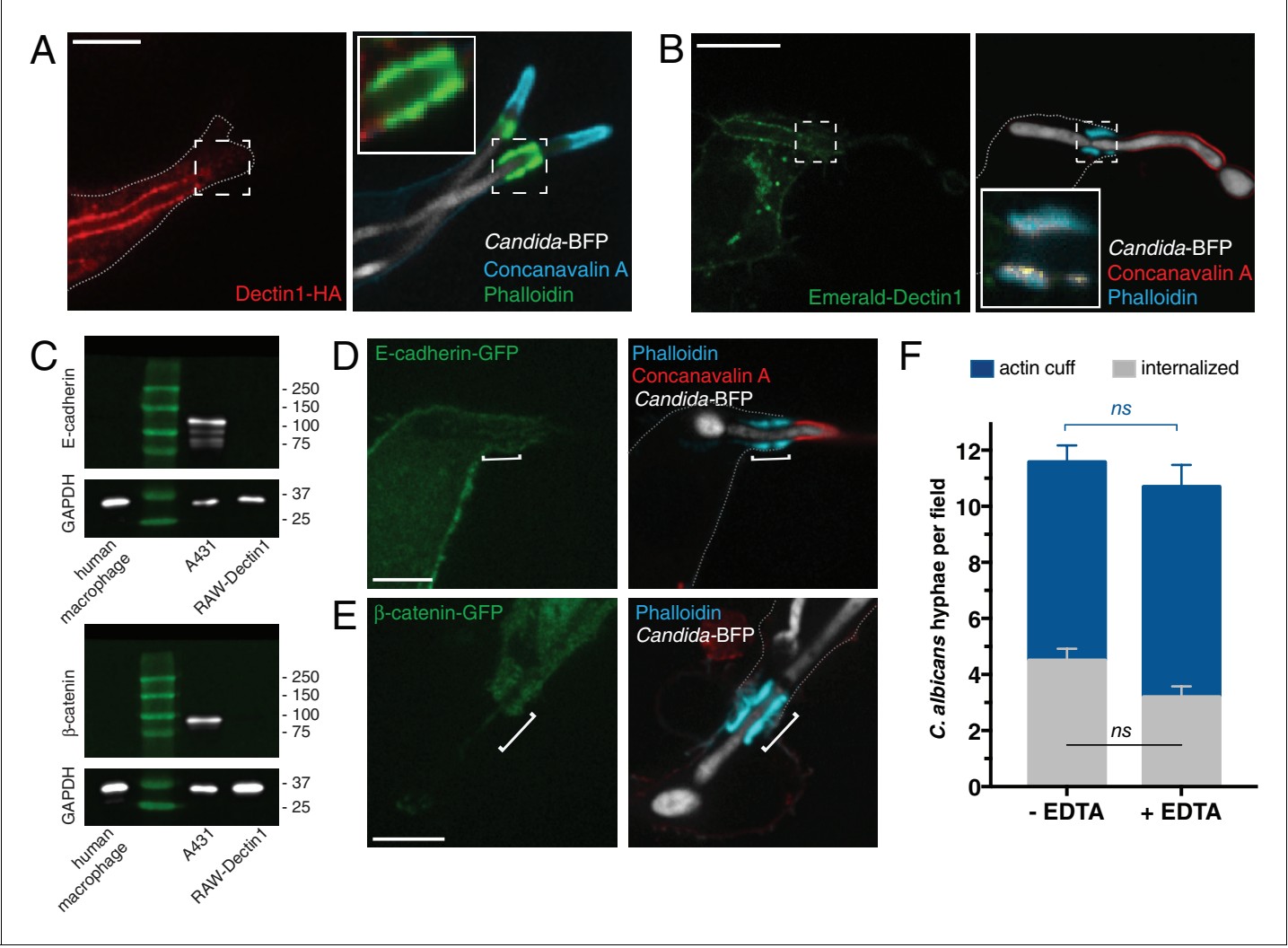

**Figure 2.** Assessing the contribution of Dectin1 and cadherin/catenin to the formation of the actin cuff. After incubation with *Candida*-BFP hyphae, RAW-Dectin1 cells were fixed and monolayers stained and visualized as follows. (**A**) The distribution of Dectin1-HA was detected by immunostaining (red). Actin was stained using fluorescent phalloidin (green); concanavalin A (blue). Inset: actin cuff shows little colocalization (yellow) with Dectin1-HA. (**B**) Visualization of Emerald-Dectin1 (green). Actin was stained using fluorescent phalloidin (blue); concanavalin A (red). Inset: poor colocalization of actin cuff with Emerald-Dectin1, in yellow. (**C**) The expression of E-cadherin (top panel) and β-catenin (bottom panel) was assessed by immunoblotting in human macrophages, A431 and RAW-Dectin1 cells; GAPDH was used as loading control. Visualization of: (**D**) E-cadherin-GFP or (**E**) β-catenin-GFP transiently transfected into RAW-Dectin1 cells. For both (**D**) and (**E**), after phagocytosis and fixation, extracellular *C. albicans* was stained using Alexa594-conjugated concanavalin A (red), and actin stained using fluorescent phalloidin (blue). Scale bars: 5 µm. (**F**) RAW-Dectin1 cells were allowed to internalize *C. albicans*-hyphae in the presence or absence of 4 mM EDTA. Following phagocytosis, extracellular *C. albicans* was stained using concanavalin A, and actin stained with phalloidin. The number of *C. albicans* hyphae that were fully internalized or partially internalized with actin cuffs per 37.5x field was counted by confocal microscopy, and the average number per field calculated. Average number of *C. albicans* per field was 12.7 ± 1.0. For each condition, three independent experiments were quantified, with ≥15 fields counted per replicate. *p* value was calculated using the unpaired, 2-tailed students t-test. Data are means ±SEM.

DOI: https://doi.org/10.7554/eLife.34798.006

The following source data and figure supplement are available for figure 2:

**Source data 1.** Numerical data corresponding to *Figure 2F*.
DOI: https://doi.org/10.7554/eLife.34798.008
**Figure supplement 1.** Cadherins accumulate at the actin cuff of epidermal cells.
DOI: https://doi.org/10.7554/eLife.34798.007

(*Figure 3—figure supplement 1*), consistent with the notion that cuff formation was independent of canonical activation of integrins, which involves Rap1 (reviewed in *Hogg et al., 2011*). There is, however, one atypical instance where integrin activation can occur in the absence of divalent cations. The α chain of the integrin complement receptor 3 (CR3, also referred to as Mac1), is unique in that it contains a lectin-like domain (LLD) that binds carbohydrates in a divalent cation-independent manner (*Thornton et al., 1996*). The LLD is separate from the I-domain –the conventional ligand-binding domain of integrins (reviewed in *Ross, 2002*)– and, interestingly, binds fungal β-glucan (*Ross et al., 1985*; *Vetvicka et al., 1996*). We therefore proceeded to test whether CR3, which consists of $\alpha_M$ (CD11b) and $\beta_2$ (CD18) subunits, is present in the region of the actin cuff. As illustrated in *Figure 3*, both CD11b and CD18 accumulated in the region of the actin cuff in RAW-Dectin1 cells that had partially internalized *C. albicans* hyphae (CD11b ratio cuff: cup 4.75 ± 0.29; n = 30, p<0.0001; CD18 ratio cuff: cup 4.79 ± 0.28; n = 30, p<0.0001; *Figure 3A,B* and insets). Moreover, talin, vinculin and paxillin were also localized to the cuff (*Figure 3C,D* and insets; *Figure 3—figure supplement 1E* and inset), as was HS1, the homologue of cortactin in leukocytes (*Figure 3E* and inset). Like cortactin, HS1 is thought to regulate actin nucleation and branching (*Daly, 2004*).

The preceding findings support a model whereby ligation of β-glucan by the LLD causes outside-in activation of CR3 directly (*O'Brien et al., 2012*; *Vetvicka et al., 1996*), or in conjunction with Dectin1 signaling (*Huang et al., 2015*; *Li et al., 2011*), resulting in Arp2/3-dependent actin nucleation. This model was tested using the M1/70 antibody, which binds to CD11b between its β-propeller and thigh domains (residues 614–682; *Osicka et al., 2015*) and effectively blocks the binding of CR3 to β-glucan (*Xia et al., 1999*). Cells pretreated with M1/70 failed to show accumulation of CR3 around partially internalized *C. albicans* hyphae, and their ability to form actin cuffs was markedly impaired (*Figure 3H*); actin cuffs were much less prominent or missing altogether when CR3 was blocked (*Figure 3F* versus G). The number of fully internalized *C. albicans* did not differ between conditions (*Figure 3H*). We concluded that binding of the CR3 integrin to *C. albicans* was critical for the establishment of long-enduring actin cuffs observed during frustrated phagocytosis of the hyphae.

## Role of receptor cooperativity in actin cuff formation

Dectin1 and CR3 both bind β-glucans (*Brown and Gordon, 2001*; *Brown et al., 2002*; *Ross et al., 1985*; *Vetvicka et al., 1996*), and have been reported to cooperate during phagocyte responses to fungal pathogens (*Huang et al., 2015*; *Li et al., 2011*). Dectin1 has also been reported to cooperate with TLR2, TLR4 (*Ferwerda et al., 2008*; *Netea et al., 2006*; *Netea et al., 2002*) and mannose receptors (*Astarie-Dequeker et al., 1999*; *Bain et al., 2014*; *Lewis et al., 2012*; *McKenzie et al., 2010*; *Netea et al., 2006*) in the recognition of *C. albicans*. We therefore sought to clarify the receptors and ligands involved in actin cuff formation.

Untransfected RAW 264.7 cells express negligible levels of Dectin1 (*Brown et al., 2003*; *Esteban et al., 2011*; *Taylor et al., 2004*), providing a means to assess the contribution of this receptor to actin cuff formation. As shown in *Figure 4A*, RAW 264.7 cells rarely formed actin cuffs compared to RAW-Dectin1 cells, suggesting that initial engagement of the hyphae by Dectin1 was essential. The requirement for Dectin1 in *C. albicans* phagocytosis (*Marakalala et al., 2013*; *Taylor et al., 2007*) could be bypassed when the hyphae were serum-opsonized, enabling opsonin receptors to establish the initial contact with the fungus (*Figure 4A*). Thus, while not accumulating in the region of the cuff, Dectin1 binding to the hyphae (which is evident by its accumulation in the frustrated phagocytic cup; *Figure 2A and B*) is required for the subsequent activation of F-actin polymerization by CR3.

We also studied cooperativity by using soluble ligands to competitively block defined receptors, and scoring the frequency of actin cuff formation (*Figure 4B*). Soluble mannan, a ligand for mannose receptor, had no effect on actin cuff formation by RAW-Dectin1 cells. Accordingly, we did not find mannose receptors in the membrane lining the actin cuff (data not shown). Laminarin, a soluble β-glucan ligand for Dectin1 (*Brown and Gordon, 2001*; *Brown et al., 2002*) impaired phagocytosis and actin cuff formation when present prior to and during phagocytosis, but not if added after the hyphae had adhered to the RAW-Dectin1 cells (*Figure 4B*). These findings support the notion that Dectin1, but not mannan receptors, cooperate with CR3 to generate the actin cuffs.

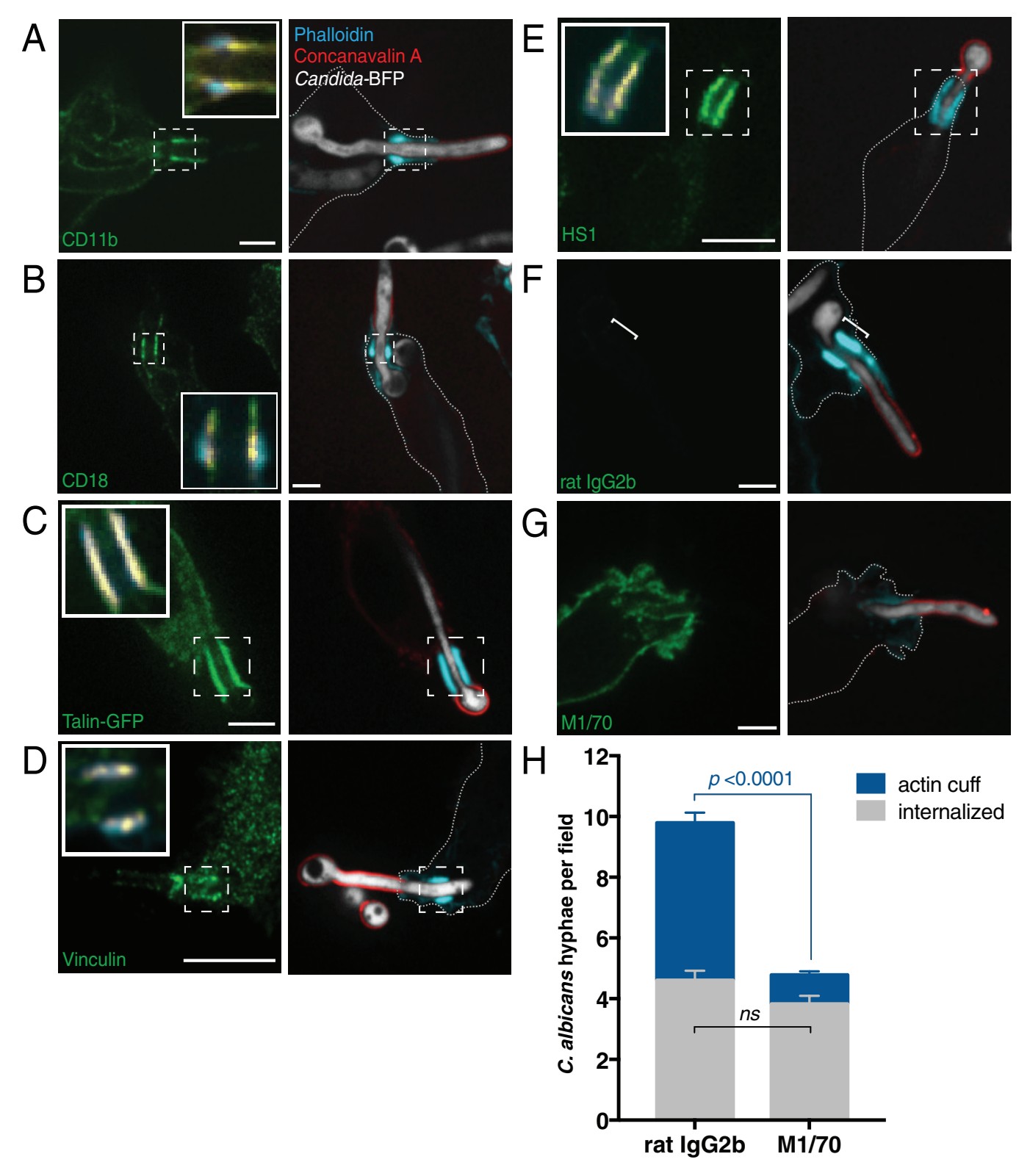

**Figure 3.** Engagement of integrin $\alpha_M\beta_2$ (CD11b/CD18) is necessary for formation of the actin cuff. After incubation with *Candida*-BFP hyphae, RAW-Dectin1 cells were fixed and extracellular *C. albicans* stained using Alexa594-conjugated concanavalin A (red). For panels (**A–G**) F-actin was stained using fluorescent phalloidin (blue), and actin cuff location indicated with a dashed box or bracket. (**A**) Anti-CD11b immunostaining (green). Inset: Colocalization of actin cuff with CD11b, in yellow. Scale bar: 5 μm. (**B**) Anti-CD18 immunostaining (green). Inset: Colocalization of actin cuff with CD18,

*Figure 3 continued on next page*

**Figure 3 continued**

in yellow. Scale bar: 5 µm. (C) Visualization of transfected Talin-GFP. Inset: Colocalization of actin cuff with talin, in yellow. Scale bar: 10 µm. (D) Immunostaining of endogenous vinculin (green). Inset: Colocalization of actin cuff with vinculin, overlaid in yellow. Scale bar: 10 µm. (E) Immunostaining of endogenous HS1 (green). Scale bar: 10 µm. (F–H) Internalization of *Candida*-BFP hyphae was allowed to proceed in the presence of the CD11b blocking antibody M1/70 or an isotype-matched (rat IgG2b) control antibody. Following phagocytosis, extracellular *C. albicans* was stained using Alexa594-conjugated concanavalin A (red), and actin stained using fluorescent phalloidin (blue). Immunostaining (green) for rat IgG2b isotype control (F, left panel) or M1/70 (G, left panel). Scale bars: 5 µm. Images shown are representative of at least 3 experiments of each kind. (H) The number of *C. albicans* hyphae that were fully internalized or partially internalized with actin cuffs per 37.5x field was counted by confocal microscopy. Average number of *C. albicans* per field was 11.7 ± 0.5. For each condition, four independent experiments were quantified, with ≥15 fields counted per replicate. *p* value was calculated using the unpaired, 2-tailed students t-test. Data are means ±SEM.

DOI: https://doi.org/10.7554/eLife.34798.009

The following source data and figure supplement are available for figure 3:

**Source data 1.** Numerical data corresponding to *Figure 3H*.

DOI: https://doi.org/10.7554/eLife.34798.011

**Figure supplement 1.** Novel activation of CR3 during actin cuff formation.

DOI: https://doi.org/10.7554/eLife.34798.010

## Fungal cell wall components that contribute to actin cuff formation

*C. albicans* cell wall components include β-(1,3)-glucans, β-(1,6) glucans, O- and N-linked mannans and chitin (*Netea et al., 2008*; *Ruiz-Herrera et al., 2006*). These can contribute to the recognition of *C. albicans* by phagocytes (reviewed in *Netea et al., 2008*), and potentially also to actin cuff formation. To clarify the contribution of individual wall components we used *g*ene *r*eplacement *a*nd *c*onditional *e*xpression (GRACE) strains (*Roemer et al., 2003*) with specific depletion targeting chitin, mannan, and β(1,6)-glucan biosynthetic pathways upon incubation with doxycycline (*Table 1*; *O'Meara et al., 2015*). Repression of pathways involved in chitin, mannan and β(1,6)-glucan synthesis using doxycycline did not affect actin cuff formation (*Figure 4C,D* and data not shown), implying that these components are dispensable. We next assessed the role of β(1,3)-glucan through pharmacological inhibition of Fks1 with caspofungin (*Douglas et al., 1997*), as genetic depletion of Fks1 results in defects in hyphae formation (*Ben-Ami et al., 2011*). Remarkably, the ability to form actin cuffs was greatly reduced in *C. albicans* grown and allowed to form hyphae in the presence of caspofungin (*Figure 4E and F*). The inhibitory effect of caspofungin on actin cuff formation was dose-dependent (*Figure 4F*), reaching ≈ 80% at 5 ng mL$^{-1}$ caspofungin, a dose that reduced the β(1,3)-glucan content of the wall by 55.3%, as assessed by aniline blue staining. Actin cuff formation around caspofungin-treated hyphae could not be rescued by serum opsonization (*Figure 4—figure supplement 1*), suggesting that β(1,3)-glucan is the ligand that promotes actin cuff assembly via CR3. Interestingly, *Aspergillus fumigatus* hyphae (routinely exceeding 80 µm in length) were also able to illicit actin cuff formation by RAW-Dectin1 cells (*Figure 4G*). *A. fumigatus* hyphae, while displaying some unique cell wall components compared to *C. albicans* hyphae, also have cell wall-associated β(1,3)-glucan (*Erwig and Gow, 2016*). We concluded that ligation of fungal β(1,3)-glucan by CR3 is required for actin cuff formation during frustrated phagocytosis of long hyphae.

## Signals driving actin cuff formation

Despite the paucity of Dectin1 and mannose receptors (*Figure 2A and B*), phosphotyrosine was markedly concentrated at the cuff (*Figure 5A*), possibly as a consequence of CR3 activation. While there is disagreement over the requirement of Src-family kinases (SFKs) for the interaction of phagocytes with fungal targets (*Elsori et al., 2011*; *Herre et al., 2004*; *Le Cabec et al., 2002*; *Mansour et al., 2013*; *Underhill et al., 2005*), there is evidence that Syk, as well as Pyk2 and Fak, two related tyrosine kinases, participate in CR3-mediated phagocytosis (*Li et al., 2006*; *Paone et al., 2016*; *Zhao et al., 2016*). The contribution of individual kinases to the tyrosine phosphorylation was explored next.

Phosphorylated SFKs accumulated along the frustrated phagocytic cup (*Figure 5A*) where Dectin1 was also found (*Figure 2A and B*), but were not particularly enriched in the region of the actin cuff (ratio cuff: cup 0.98 ± 0.03; n = 17, p=0.61). SFK inhibition by PP2 following adherence of the hyphae to RAW-Dectin1 cells had no effect on actin cuff formation (*Figure 5E*). In contrast, the phosphorylated (active) forms of Pyk2 and Fak were enriched solely at the actin cuff (pPyk2 ratio cuff: cup

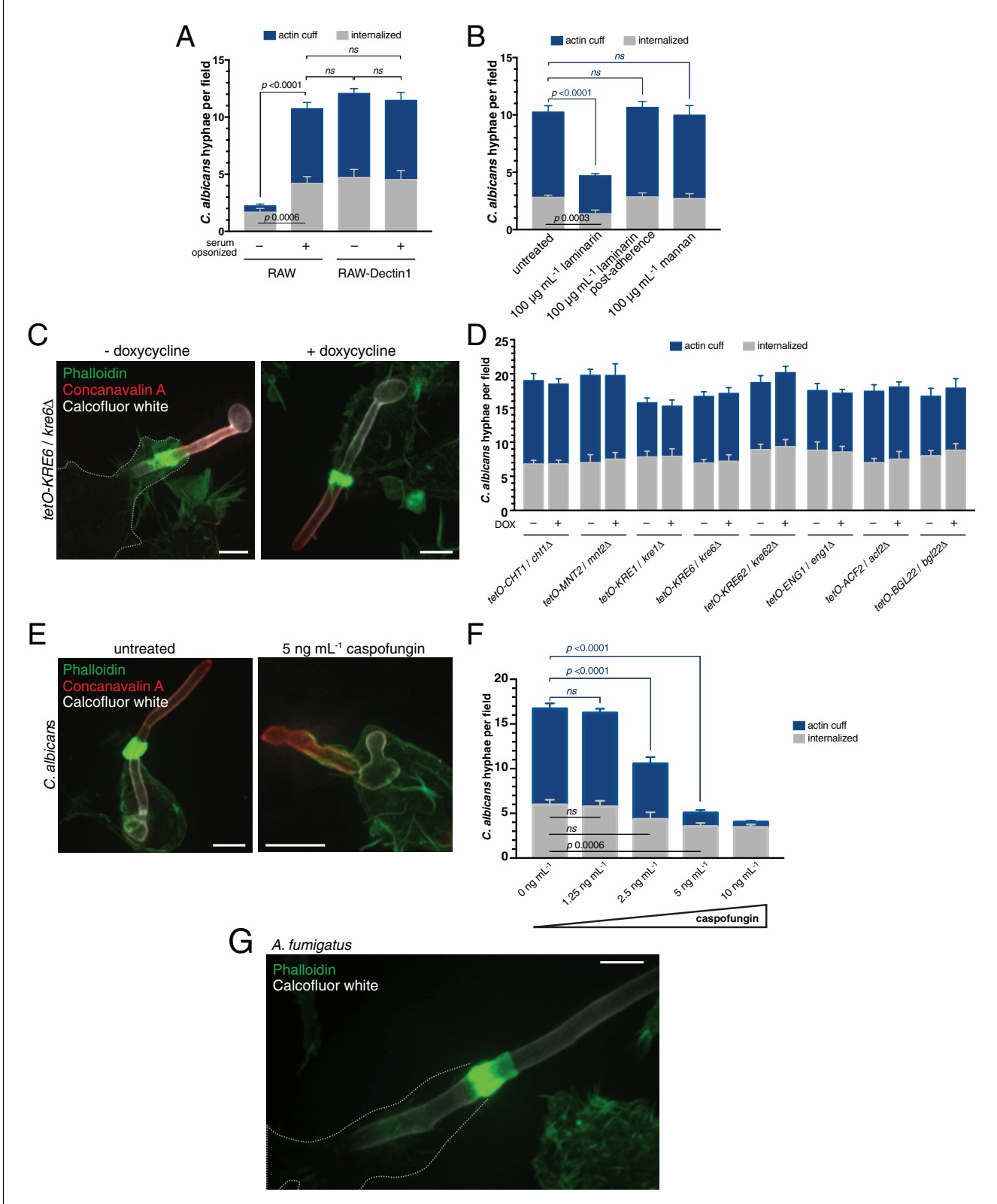

**Figure 4.** Assessing the contribution of C. albicans cell wall components to actin cuff formation. (**A**) RAW or RAW-Dectin1 cells were incubated with *Candida*-BFP hyphae that had been either untreated or serum-opsonized. Following phagocytosis, extracellular *C. albicans* was stained using concanavalin A, and actin stained with phalloidin. The number of *C. albicans* hyphae that were fully internalized or partially internalized with actin cuffs per 37.5x field was counted by confocal microscopy. Average number of *C. albicans* per field was 15.7 ± 1.3. For each condition, three independent

*Figure 4 continued on next page*

*Figure 4 continued*

experiments were quantified, with ≥4 fields counted per replicate. *p* value was calculated using the unpaired, 2-tailed students t-test. Data are means ±SEM. (B) RAW-Dectin1 cells were allowed to internalize *Candida*-BFP hyphae in the presence or absence of laminarin or mannan. For laminarin, RAW-Dectin1 cells were also allowed to adhere *C. albicans* 15 min prior to the addition of laminarin, as indicated. Other details as in A. Average number of *C. albicans* per field was 12.9 ± 0.7. (C–D) Evaluation of *C. albicans* GRACE strain cell wall mutants for actin cuff formation. GRACE strains were induced to form hyphae in the absence or presence of doxycycline (DOX) to repress target gene expression, and incubated with RAW-Dectin1 cells. Following phagocytosis, monolayers were fixed and *C. albicans* stained with 10 μg mL$^{-1}$ calcofluor white (white), extracellular *C. albicans* stained using concanavalin A (red), and actin stained with phalloidin (green). Image in C is representative of ≥30 fields from ≥3 separate experiments of each type. Scale bar: 5 μm. (D) The number of *C. albicans* hyphae that were fully internalized or partially internalized with actin cuffs per 37.5x field was counted by confocal microscopy, and the average number per field calculated. Average number of *C. albicans* per field was 20.6 ± 0.6. For each condition, three independent experiments were quantified, with ≥4 fields counted per replicate. *p* value was calculated using the unpaired, 2-tailed students t-test. Data are means ±SEM. (E) Role of *C. albicans* β-(1,3)-glucan in actin cuff formation. The GRACE wild-type strain was incubated and induced to form hyphae in the presence or absence of 5 ng mL$^{-1}$ caspofungin and incubated with RAW-Dectin1 cells for phagocytosis. Following phagocytosis, cells were fixed and *C. albicans* stained with 10 μg mL$^{-1}$ calcofluor white (white), extracellular *C. albicans* stained using fluorescent concanavalin A (red), and actin stained with fluorescent phalloidin (green). Image is representative of ≥30 fields from ≥3 separate experiments. Scale bar: 5 μm. (F) The effect of β-(1,3)-glucan synthase inhibition on actin cuff formation. Hyphae were prepared as in (E), with varying concentrations of caspofungin, as indicated. Phagocytosis, fixation and staining as in (E). Other details as in (A). Average number of *C. albicans* per field was 19.7 ± 0.8. (G) Actin cuffs are observed during phagocytosis of *A. fumigatus* hyphae. After incubation with hyphae, monolayers were fixed and *A. fumigatus* stained with 10 μg mL$^{-1}$ calcofluor white (white). Actin stained with phalloidin (green). Image representative of ≥30 fields from ≥2 separate experiments. Scale bar: 5 μm.

DOI: https://doi.org/10.7554/eLife.34798.012

The following source data and figure supplement are available for figure 4:

**Source data 1.** Numerical data corresponding to *Figure 4A*.
DOI: https://doi.org/10.7554/eLife.34798.014

**Source data 2.** Numerical data corresponding to *Figure 4B*.
DOI: https://doi.org/10.7554/eLife.34798.015

**Source data 3.** Numerical data corresponding to *Figure 4D*.
DOI: https://doi.org/10.7554/eLife.34798.016

**Source data 4.** Numerical data corresponding to *Figure 4F*.
DOI: https://doi.org/10.7554/eLife.34798.017

**Figure supplement 1.** *C. albicans* β-(1,3)-glucan is required for actin cuff formation.
DOI: https://doi.org/10.7554/eLife.34798.013

23.69 ± 1.20; n = 46, p<0.0001, pFak ratio cuff: cup 22.56 ± 1.01; n = 34, p<0.0001; *Figure 5C and D*). Moreover, inhibition of Pyk2/Fak activity by PF573228 following adherence of the hyphae to the cells abolished actin cuff formation, with no effect on internalization (*Figure 5E*). Also, as reported by *Strijbis et al., 2013*, we observed phosphorylation of Syk with accumulation at the actin cuff (ratio cuff: cup 21.55 ± 1.75; n = 22, p<0.0001; *Figure 5—figure supplement 1*). As expected, inhibition of Syk by piceatannol after *C. albicans* adherence blocked actin cuff formation (*Figure 5E*). These data provide evidence that, along with Syk, Pyk2/Fak play a role in the interaction between macrophages and *C. albicans*, and are important for actin cuff formation during frustrated phagocytosis of hyphae.

Interestingly, the interaction of Pyk2 with β$_2$ integrins activates Vav1 (*Gakidis et al., 2004*; *Kamen et al., 2011*), a GEF for Rho-family GTPases that is also essential for the phagocytosis and control of *C. albicans* by macrophages (*Strijbis et al., 2013*). Accordingly, Rac1 and/or Cdc42 were seemingly involved in the marked polymerization of actin at the cuff. This was indicated by the recruitment of PAK(PBD), a biosensor of the active (GTP-bound) form of these GTPases (*Benard et al., 1999*), that accumulated at the cuff to levels ≥4 fold higher than along the cup. F-actin accumulation at the cuff was sensitive to the formin inhibitor SMI-FH2, but not to the Arp2/3 inhibitor CK-666 (*Figure 5G*). Together, these data suggest that activation of Syk and Pyk2/Fak by CR3 leads to activation of Rho-family GTPases, culminating in formin-mediated actin assembly, a process akin to focal adhesion formation (reviewed in *Vicente-Manzanares et al., 2005*).

**Table 1.** *C. albicans* strains used in this study.

| Strain | Parent | Genotype | Gene function | Reference |
|---|---|---|---|---|
| Candida-BFP | SC5314 | $P_{eno1}$-TagBFP-NAT$^R$ | N/A | (*Strijbis et al., 2013*) |
| CaSS1 | CAI4 | ura3::imm$^{434}$ / ura3::imm$^{434}$ his3::hisG / his3::hisG leu2::tetR-GAL4AD-URA / LEU2 | N/A | (*Roemer et al., 2003*) |
| CHT1 | CaSS1 | tetO-CHT1 / cht1Δ | chitinase | (*O'Meara et al., 2015*) |
| CDA2 | CaSS1 | tetO-CDA2 / cda2Δ | chitin deacetylase | (*O'Meara et al., 2015*) |
| MNT2 | CaSS1 | tetO-MNT2 / mnt2Δ | α-(1,2)-mannosyl transferas | (*O'Meara et al., 2015*) |
| VRG4 | CaSS1 | tetO-VRG4 / vrg4Δ | GDP-mannose transporte | (*O'Meara et al., 2015*) |
| KRE1 | CaSS1 | tetO-KRE1 / kre1Δ | cell wall glycoprotein, β-(1,6)-glucan synthesis | (*O'Meara et al., 2015*) |
| KRE6 | CaSS1 | tetO-KRE6 / kre6Δ | β-(1,6)-glucan synthase subunit | (*O'Meara et al., 2015*) |
| KRE62 | CaSS1 | tetO-KRE62 / kre62Δ | β-(1,6)-glucan synthase subunit | (*O'Meara et al., 2015*) |
| KEG1 | CaSS1 | tetO-KEG1 / keg1Δ | integral membrane ER protein, β-(1,6)-glucan synthesis | (*O'Meara et al., 2015*) |
| ENG1 | CaSS1 | tetO-ENG1 / eng1Δ | endo-(1,3)-β-glucanase | (*O'Meara et al., 2015*) |
| ACF2 | CaSS1 | tetO-ACF2 / acf2Δ | endo-(1,3)-β-glucanase | (*O'Meara et al., 2015*) |
| BGL22 | CaSS1 | tetO-BGL22 / bgl22Δ | putative β-glucanase | (*O'Meara et al., 2015*) |

DOI: https://doi.org/10.7554/eLife.34798.018

## Phospholipid segregation between the plasma membrane and the cuff-delimited phagosomal cup

Phospholipids undergo striking changes during the course of conventional phagocytosis. PtdIns(4,5)$P_2$ that is normally found in the plasma membrane is converted to PtdIns(3,4,5)$P_3$ at sites of receptor engagement, and is subsequently degraded by lipases and phosphatases, becoming undetectable in sealed phagosomes. PtdIns(3,4,5)$P_3$ can be detected for up to a minute following sealing, but then disappears abruptly as PtdIns(3)P appears; the latter is detectable on early phagosomes for about 10–15 min (reviewed in *Levin et al., 2015*). These drastic switches are thought to reflect and possibly dictate the identity and developmental stage of the maturing phagosome. It has been observed that the frustrated tubular phagosomes of heat-killed filamentous *Legionella pneumophila* are accompanied by a sharp separation of plasmalemmal and phagosomal phosphoinositide species (*Naufer et al., 2018*; *Prashar et al., 2013*). Additionally, atypical phosphoinositide dynamics can occur in sealed phagosomes containing filamenting *C. albicans* (*Heinsbroek et al., 2009*) or during CR3-mediated phagocytosis of opsonized targets (*Bohdanowicz et al., 2010*). Therefore we analyzed the phosphoinositides in frustrated phagosomes of *C. albicans* hyphae. We used the genetically-encoded fluorescent biosensor PLCδ-PH-GFP to monitor the distribution of PtdIns(4,5)$P_2$. Remarkably, while PtdIns(4,5)$P_2$ was present as expected in the surface membrane facing the extracellular milieu, it was undetectable in the invaginated section that constituted the frustrated phagosome (*Figure 6A*). In stark contrast, PtdIns(3,4,5)$P_3$ –which was visualized using AKT-PH-GFP– was found solely in the open phagosomal cup (*Figure 6B*), where it co-existed with PtdIns(3)P, detected using the PX-GFP sensor (*Figure 6C*). In addition to the localization of PtdIns(3,4,5)$P_3$ in the cup reported in a previous collaborative study (*Strijbis et al., 2013*), we detected additional enrichment of PtdIns(3,4,5)$P_3$ in the actin cuff region (ratio cuff: cup $1.39 \pm 0.10$; n = 30, p=0.0006). In contrast, PtdIns(3)P was comparatively excluded from the actin cuff (ratio cuff: cup $0.823 \pm 0.05$; n = 30,

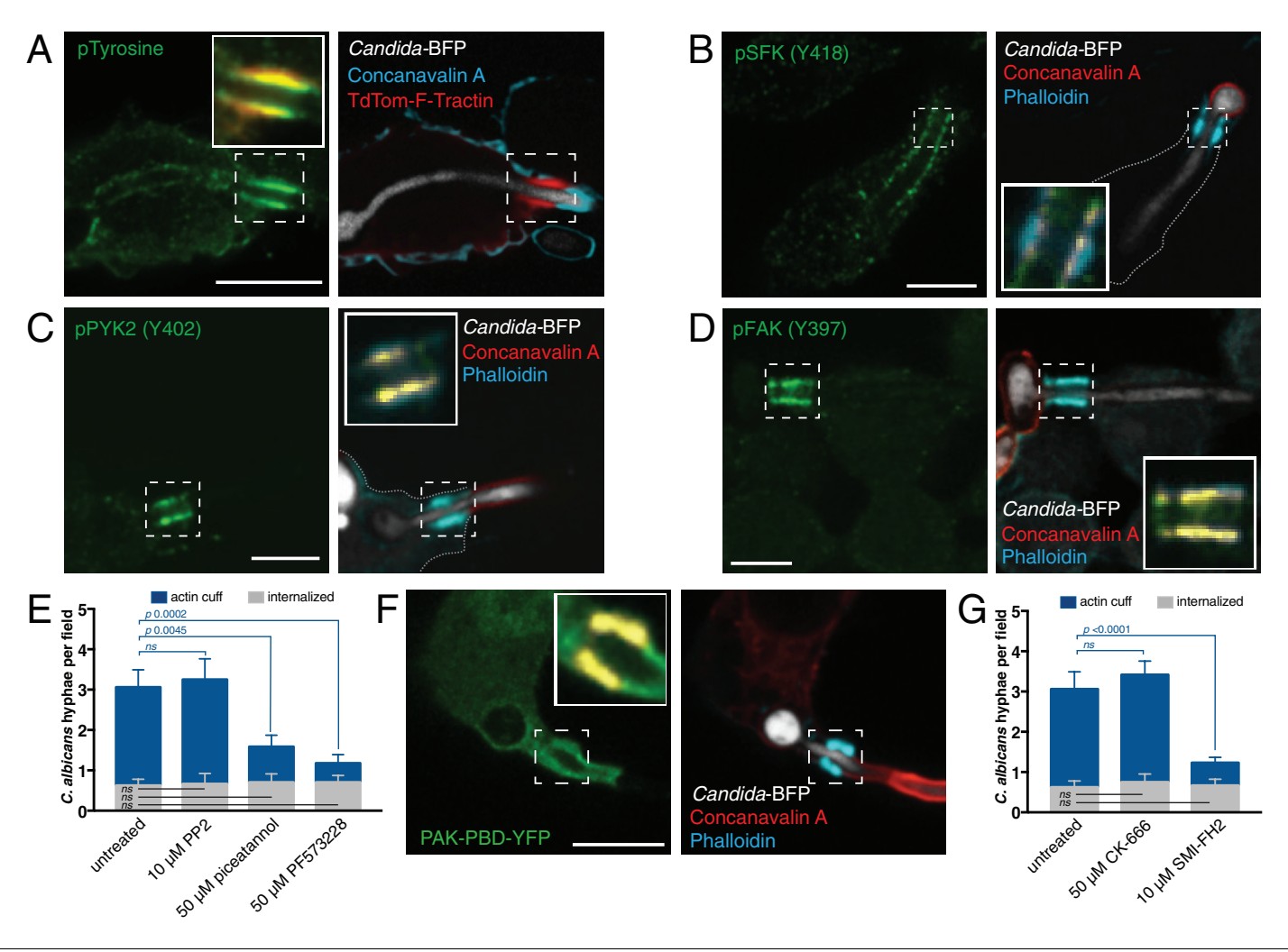

**Figure 5.** Signaling associated with actin polymerization at the phagocytic cup formed around *C. albicans* hyphae. After incubation with *Candida*-BFP hyphae, RAW-Dectin1 cells were fixed and extracellular *C. albicans* stained using fluorescent concanavalin A. (**A**) Phosphotyrosine (pTyrosine) was detected by immunostaining (green). F-actin was visualized using TdTom-F-Tractin (red); concanavalin A (blue). Inset: Colocalization of actin cuff with pTyrosine, in yellow. Image is representative of ≥30 fields from ≥3 separate experiments. (**B**) Phospho-SFK (Y418) was detected by immunostaining (green); concanavalin A (red). Inset: Colocalization of actin cuff with pSFK, in yellow. (**C**) Phospho-PYK2 (Y402) was detected by immunostaining (green); concanavalin A (red). Inset: Colocalization of actin cuff with pPYK2, in yellow. (**D**) Phospho-FAK (Y397) was detected by immunostaining (green); concanavalin A (red). Inset: Colocalization of actin cuff with pFAK, in yellow. Images in B, C and D are representative of ≥30 fields from ≥2 separate experiments of each type. (**E**) Effect of tyrosine kinase inhibitors on actin cuff formation. RAW-Dectin1 cells were allowed to adhere *Candida*-BFP hyphae for 15 min and then incubated 45 min in the presence of vehicle, PP2, piceatannol or PF573228. Following phagocytosis, extracellular *C. albicans* was stained using concanavalin A, and actin stained with phalloidin. The number of *C. albicans* hyphae that were fully internalized or partially internalized with actin cuffs per 94.5x field was counted by confocal microscopy. Average number of *C. albicans* per field was $3.4 \pm 0.6$. For each condition, three independent experiments were quantified, with ≥4 fields counted per replicate. *p* value was calculated using unpaired, 2-tailed students t-test. Data are means ±SEM. (**F**) Active Rac/Cdc42 were visualized using PAK(PBD)-YFP as a probe (green). Actin was stained using fluorescent phalloidin (blue); concanavalin A (red). Inset: Colocalization of actin cuff with PAK(PBD), in yellow. Image is representative of ≥30 fields from ≥3 separate experiments. Scale bars: 10 µm. (**G**) Effect of actin assembly inhibitors on actin cuff formation. RAW-Dectin1 cells were allowed to adhere *Candida*-BFP hyphae for 15 min, then incubated 45 min in the presence of vehicle, CK-666 or SMI-FH2. Following phagocytosis, extracellular *C. albicans* was stained using concanavalin A, and actin stained with phalloidin. The number of *C. albicans* hyphae that were fully internalized or partially internalized with actin cuffs per 94.5x field was counted by confocal microscopy. Average number of *C. albicans* per field as in (E). For each condition, three independent experiments were quantified, with ≥5 fields counted per replicate. *p* value was calculated using the unpaired, 2-tailed students t-test. Data are means ±SEM.

DOI: https://doi.org/10.7554/eLife.34798.019

The following source data and figure supplement are available for figure 5:

*Figure 5 continued on next page*

*Figure 5 continued*

**Source data 1.** Numerical data corresponding to *Figure 5E*.
DOI: https://doi.org/10.7554/eLife.34798.021
**Source data 2.** Numerical data corresponding to *Figure 5G*.
DOI: https://doi.org/10.7554/eLife.34798.022
**Figure supplement 1.** Localization of pSYK to the actin cuff.
DOI: https://doi.org/10.7554/eLife.34798.020

p=0.0025). The segregation of these phosphoinositides persisted for the duration of our observations (up to 90 min after frustrated phagosome formation; not illustrated).

## The actin cuff forms a diffusional barrier to the movement of proteins and lipids

The sharp boundary between the PtdIns(4,5)$P_2$-rich surface membrane and the tubular membrane endowed with PtdIns(3,4,5)$P_3$ and PtdIns(3)P coincided with the location of the actin cuff, suggesting that the latter may function as a diffusion barrier. However, the restricted localization of the phosphoinositides may have resulted from the strategic positioning of synthetic (i.e. kinases) and degradative (i.e. phosphatases or lipases) enzymes. To more definitively assess the existence of a diffusion barrier, we analyzed the distribution and dynamics of molecules that do not undergo rapid metabolic transformation, including lipid-anchored and transmembrane proteins, which had been reported to segregate in frustrated phagosomes. As shown in *Figure 6D*, LC3 –a small protein covalently linked to PtdEth– was found in the frustrated phagosome (*Kanayama and Shinohara, 2016*; *Martinez et al., 2015*; *Sprenkeler et al., 2016*; *Tam et al., 2016*), yet did not reach the surface membrane. Similarly, both wild-type Rab7 (*Figure 6E*) and constitutively-active Rab7 (not illustrated) are confined to the frustrated phagosomal tube and partially excluded from the actin cuff (Rab7 ratio cuff: cup $0.68 \pm 0.05$; n = 30, p<0.0001), as was LAMP1 (ratio cuff: cup $0.59 \pm 0.03$; n = 30, p<0.0001; *Figure 6F*), a late-endosomal/lysosomal membrane-spanning glycoprotein. The exclusion from the actin cuff was better appreciated by 3D visualization of LAMP1 (*Figure 6G,H* and *Video 2*). Because metabolic conversion to other species could not account for the segregation of the latter probes to the invaginated section of the membrane, we considered it more likely that restricted diffusion accounted for the observations.

It was nevertheless possible that molecules like LC3, Rab7 or LAMP were inserted through fusion into the tubular part of the membrane, where they could conceivably remain immobile. To exclude this possibility, we assessed their mobility measuring fluorescence recovery after photobleaching (FRAP). The constitutively-active form of Rab7, Rab7(Q67L), was used for these experiments; because this variant is unable to exchange nucleotides, it does not associate stably with GDI and remains membrane associated (*Méresse et al., 1995*), eliminating the confounding effects of fluorescence recovery from a cytosolic pool. Rapid recovery was observed following photobleaching of a $\approx 3$ μm spot within the phagosomal cup. In four independent experiments, half-maximal recovery was attained after 3.3 s (*Figure 7B*). Similar analyses were performed using GFP-tagged LAMP1 (*Figure 7A,B*), which also recovered within seconds ($t_{1/2}$ = 7.9 sec). Between 75–80% of the fluorescence was recovered in both instances, implying that the majority of the Rab7(Q67L) and LAMP1 molecules were mobile.

The retention of Rab7(Q67L) and LAMP1 in the cup for many minutes despite their ability to move laterally in the plane of the membrane implies that they are unable to cross the junction with the surface membrane. The existence of a diffusion barrier was confirmed by expressing the N-terminal domain of Lyn (Lyn$_{11}$) tagged with GFP. This region of the protein becomes myristoylated and palmitoylated, targeting it to the plasma membrane and, to a lesser extent, to early endosomes. Following frustrated phagocytosis of hyphae, Lyn$_{11}$-GFP is found both at the membrane and in the phagosomal cup, where its density is lower, likely because of dilution caused by insertion of unlabeled endomembranes. We analyzed comparatively small phagosomes to enable photobleaching of Lyn$_{11}$-GFP in the entire cup (*Figure 7C*). Strikingly, the fluorescence of the cup failed to recover, despite the persistence of abundant Lyn$_{11}$-GFP in the adjacent plasmalemma. In three independent experiments only 19% of the original fluorescence reappeared, possibly via fusion with Lyn$_{11}$-GFP-containing early endosomes. Failure to recover was not attributable to immobility of Lyn$_{11}$-GFP in

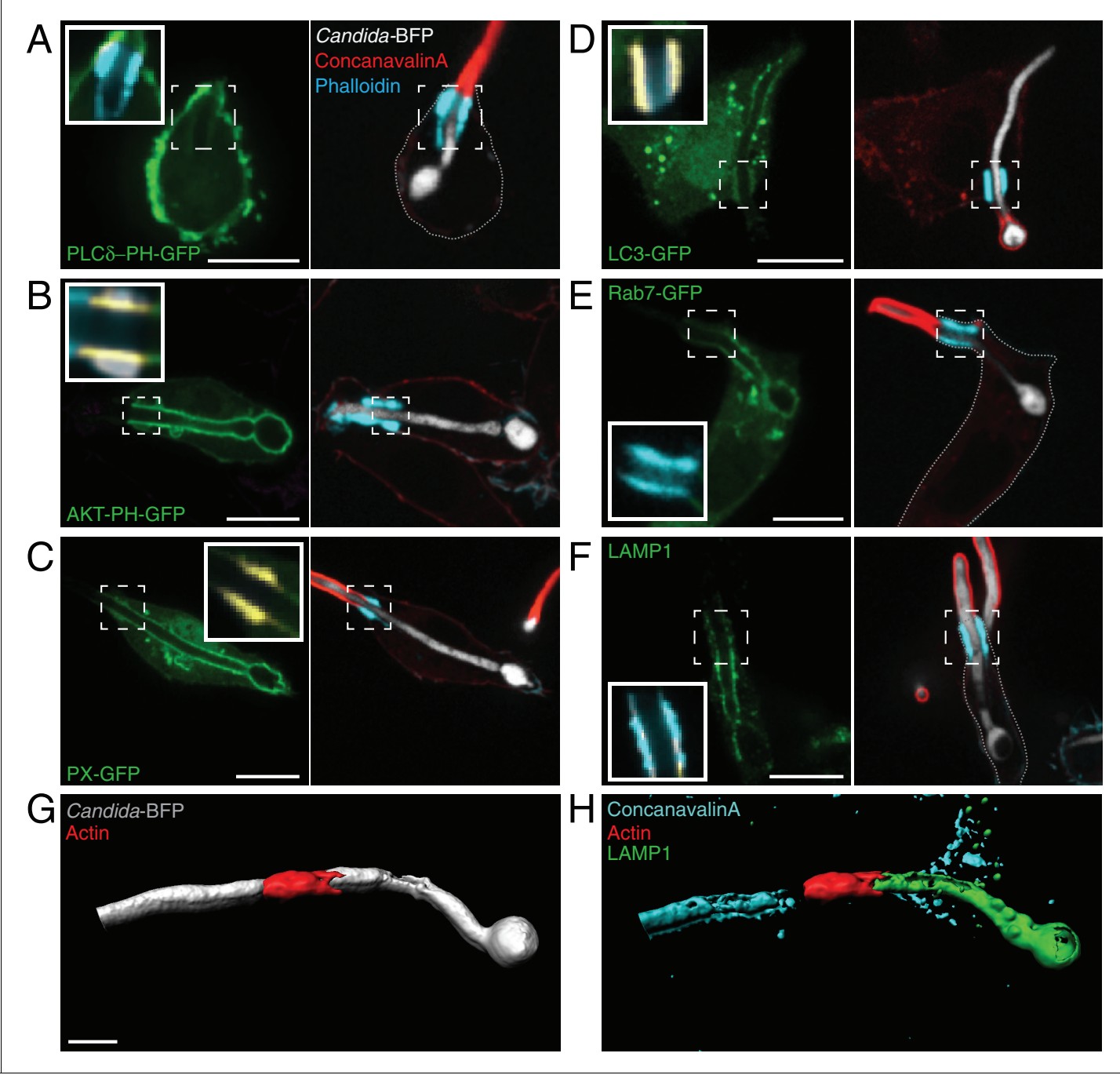

**Figure 6.** Distribution of phosphoinositides and endo-lysosomal markers. After incubation with *Candida*-BFP hyphae, RAW-Dectin1 cells were fixed and extracellular *C. albicans* stained using Alexa594-conjugated concanavalin A (red). Actin was stained using fluorescent phalloidin (blue), and the location of the actin cuff is indicated by the dashed square. Visualization of: (**A**) PtdIns(4,5)P$_2$ using PLCδ-PH-GFP; (**B**) PtdIns(3,4,5)P$_3$/PtdIns(3,4)P$_2$ using AKT-PH-GFP, inset: colocalization of actin cuff with AKT-PH, in yellow; (**C**) PtdIns(3)P using PX-GFP, inset: colocalization of actin cuff with PX, in yellow; (**D**) LC3-GFP, inset: colocalization of actin cuff with LC3, in yellow; (**E**) Rab7-GFP, inset: colocalization of actin cuff with Rab7, in yellow; (**F**) immunostained LAMP1 (green), inset: colocalization of actin cuff with LAMP1, in yellow. Scale bars: 10 μm. Images are representative of ≥30 fields from ≥3 separate experiments of each type. (**G–H**) 3D rendering of a RAW-Dectin1 cell with a partially internalized *C. albicans* hypha. After incubation with *Candida*-BFP, RAW-Dectin1 cells were fixed and extracellular portions of the hyphae were stained using concanavalin A (blue). (**G**) *C. albicans* (white) visualized with actin immunostaining (red). (**H**) Same 3D rendering as in (**G**), visualizing LAMP1 immunostaining (green), actin (red) and concanavalin A (blue). Scale bar: 5 μm.

DOI: https://doi.org/10.7554/eLife.34798.023

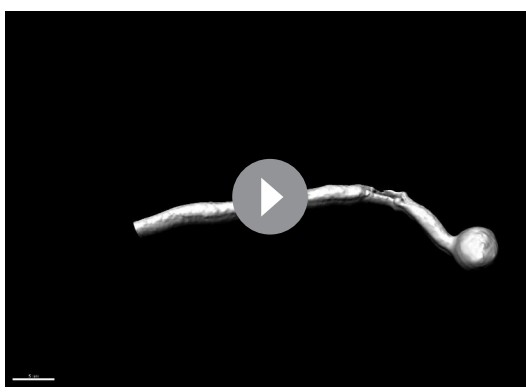

**Video 2.** 3D rendering of a RAW-Dectin1 cell with a partially internalized *Candida*-BFP hypha (white), showing the demarcation of phagosomal LAMP1 (green) and concanavalin A (blue) by the actin cuff (red). See *Figure 6* for additional information.
DOI: https://doi.org/10.7554/eLife.34798.024

the membrane, which displayed very fast and nearly complete recovery following photobleaching (*Figure 7D,E*). These data confirm that the region of the actin cuff acts as a lateral diffusion barrier, separating the inner leaflet of the plasma membrane from that of the open phagocytic cup.

It is noteworthy that while the barrier curtails the diffusion of lipids and proteins anchored to lipids on the inner leaflet of the membrane, exofacial lipids and lipid-associated proteins readily traverse the junction between the membrane and the tubular phagosome. This was demonstrated by incorporation of rhodamine-labeled PtdEth to the surface membrane following stabilization of the frustrated phagosome. The labeled lipid, which inserts into the outer leaflet of the plasmalemma, reached the entire membrane of the frustrated phagocytic cup within ≈5 min (*Figure 7—figure supplement 1A*). Similarly, fluorescent cholera toxin B subunit, which binds to exofacial ganglioside $GM_1$, promptly entered the phagocytic cup (*Figure 7—figure supplement 1B*). Thus, the actin-dependent diffusion barrier selectively restricted the mobility of components of the inner leaflet, including transmembrane proteins, while exofacial lipids remained able to traverse the junction.

## Examining the role of CR3 and actin in the maintenance of diffusional barriers

How is the diffusion barrier generated? We speculated that the molecular crowding resulting from tight clustering of integrins and their ancillary proteins could restrict the diffusion of membrane-associated components across the cuff. To test this possibility, we investigated whether sufficient molecular crowding could be generated to exclude other membrane components from regions of integrin clustering. To this end, we used antibody-induced cross-linking, a strategy shown earlier to induce the formation of CR3 patches on the plasma membrane (*Fukushima et al., 1996*; *Pavan et al., 1992*; *Zhou et al., 1993*). Whether exclusion could be induced by molecular crowding was assessed analyzing the distribution of CD2-CD45-GFP (*Figure 8B*), a transmembrane protein having a short, 7 nm ectodomain (*Cordoba et al., 2013*). As shown in *Figure 8A*, prior to cross-linking both CD2-CD45-GFP and CR3 were distributed diffusely throughout the membrane, overlapping extensively at the resolution of the confocal microscope. The CD2-CD45-GFP fluorescence intensity in CR3-positive regions compared to the average CD2-CD45-GFP fluorescence intensity of the entire plasma membrane averaged 0.69 ± 0.01 (585 CR3-positive regions in 20 cells from three different experiments). After antibody treatment, CR3 clustered into large, dense patches. Strikingly, CD2-CD45-GFP was largely (81%) and significantly (p>0.0001) excluded from such patches, where the fluorescence was only 0.13 ± 0.01 of the plasmalemmal average (measured in 472 CR3 patches in 15 cells from three experiments). Importantly, the exclusion was not alleviated by treatment with latrunculin A, the fluorescence of the patches averaging 0.12 ± 0.01 of the plasmalemmal average (measured in 445 patches in 18 cells from three experiments), implying that the actin cytoskeleton is not involved in the domain segregation. CD2-CD45-GFP exclusion did not differ between these two conditions (p=0.66). We concluded that integrins could be sufficiently clustered to exclude other membrane components. By forming a continuous and thick ring around the neck of the frustrated phagosome, the molecular crowding of clustered integrins could generate a diffusional barrier.

While actin is not essential to constrain the diffusion across patches of antibody-aggregated integrins, it is nevertheless required to maintain the integrins clustered in response to the glucan during frustrated phagocytosis. As such, an intact actin cuff is required to establish and maintain the barrier to phosphoinositides or transmembrane proteins. This was validated in cells that had formed a stable frustrated phagosome around *C. albicans* hyphae and were then treated with latrunculin A,

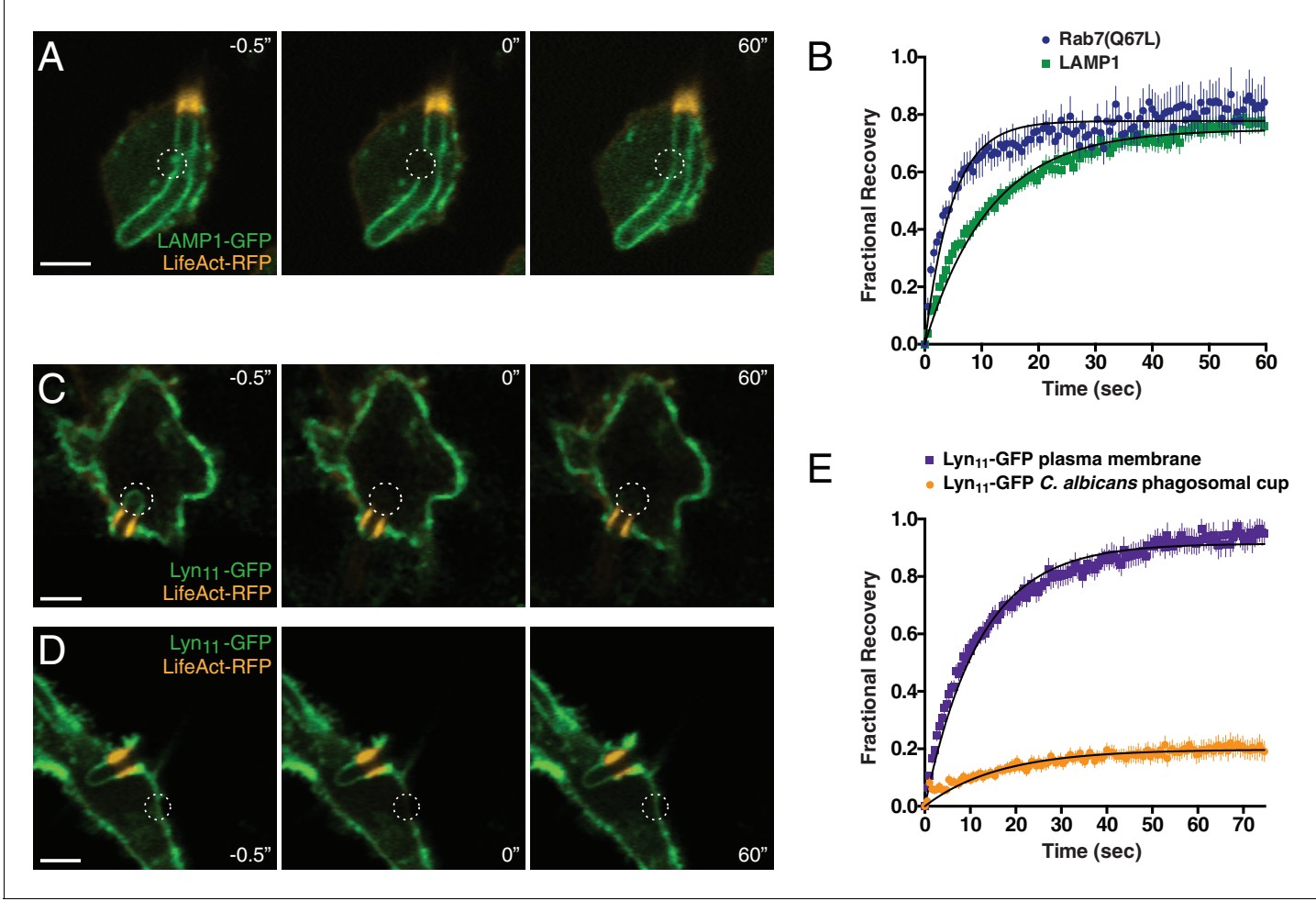

**Figure 7.** Formation of the actin cuff is associated with the establishment of a diffusional barrier. RAW-Dectin1 cells were transfected with the indicated constructs, exposed to *Candida*-BFP, and used for FRAP determinations. F-actin was visualized with LifeAct-RFP (orange). (A) A region of interest (denoted by dotted circle) of LAMP1-GFP in the frustrated phagosome was selected (left panel, −0.5"), photobleached (middle panel, 0"), and allowed to recover for 60 s (right panel, 60"). Scale bar: 5 μm. Images in A, C and D are representative of ≥30 fields from ≥3 separate experiments of each type. (B) Quantitation of fractional recovery of fluorescence after photobleaching LAMP1 (green) or Rab7(Q67L) (blue). In both cases, data were normalized to fluorescence in unbleached regions of the *C. albicans* phagosomal cup. For either condition, four biological replicates, with a total of ≥30 cells, were quantified. (C–D) A region of interest in the frustrated phagosome (C) or in the plasma membrane (D) of cells expressing Lyn$_{11}$-GFP was selected (left panel, −0.5"), photobleached (middle panel, 0"), and allowed to recover for 60 s (right panel, 60"). Scale bars: 5 μm. (E) Quantitation of fractional recovery of fluorescence of photobleached Lyn$_{11}$-GFP in the plasma membrane (blue) or the frustrated *C. albicans* phagosomal cup (orange). In both cases, FRAP data was normalized to fluorescence in the plasma membrane. For either condition, three biological replicates, with a total of ≥35 cells, were quantified.

DOI: https://doi.org/10.7554/eLife.34798.025

The following source data and figure supplement are available for figure 7:

**Source data 1.** Numerical data corresponding to *Figure 7B*.
DOI: https://doi.org/10.7554/eLife.34798.027
**Source data 2.** Numerical data corresponding to *Figure 7E*.
DOI: https://doi.org/10.7554/eLife.34798.028
**Figure supplement 1.** Diffusion of outer leaflet components is not restricted by the actin cuff.
DOI: https://doi.org/10.7554/eLife.34798.026

which was shown earlier (*Figure 1I*) to cause gradual disassembly of the cuff. PtdIns(4,5)P$_2$ –which in untreated cells is excluded from the phagocytic cup (*Figures 6A* and *9A*)– gained access to the entire cup when actin was disassembled by latrunculin (*Figure 9B*). The PtdIns(4,5)P$_2$ present in the cup, expressed relative to the plasmalemma, increased 4.88 times after latrunculin treatment

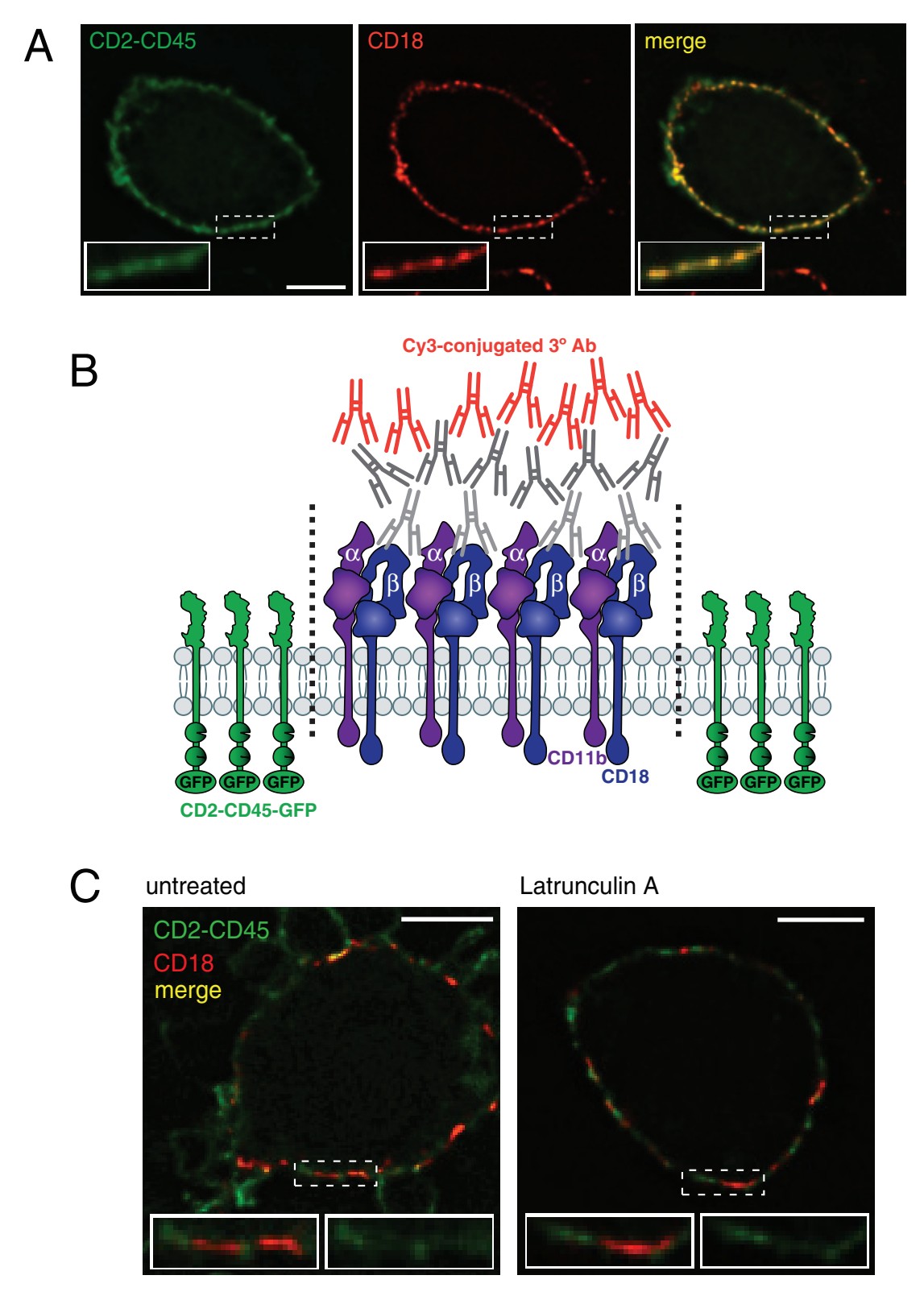

**Figure 8.** Clustering and patching of CR3 forms a diffusional barrier that excludes transmembrane proteins. (**A**) Raw-Dectin1 cells transiently expressing CD2-CD45-GFP (left panel, green) and stained for external CD18 (middle panel, red). Right panel shows the colocalization of CD2-CD45 and CD18, in yellow. Panel insets: 2.1x magnification. Scale bar: 5 μm. (**B**) Diagram illustrating the method used to cluster CR3 in CD2-CD45-GFP-expressing Raw-Dectin1 cells, using M18/2 antibody to CD18, followed by secondary and tertiary antibodies. See Materials and methods for details. (**C**) Effect of CR3

*Figure 8 continued on next page*

*Figure 8 continued*

patching and actin depolymerization. After clustering CR3 as in (B), cells were incubated 10 min in the absence (left) or presence (right) of 1 μM latrunculin A, and visualized for CD2-CD45-GFP (green) and extracellular CD18 (red). Colocalization of CD2-CD45 and CD18 channels shown in yellow. Panel insets: 2x magnification of both channels (left inset) and CD2-CD45 channel only (right inset). Scale bars: 5 μm.

DOI: https://doi.org/10.7554/eLife.34798.029

(*Figure 9C*). Conversely, LAMP1 –that is restricted to the cup in untreated cells (*Figures 6F* and *9D*)– was able to reach the surface membrane following treatment with latrunculin (*Figure 9E*). After latrunculin treatment, the ratio of LAMP1 present in the cup decreased 3.99 times (*Figure 9F*). Clearly, while clustering of CR3 is sufficient to form a diffusional barrier (*Figure 8*), the actin cuff formed during phagocytosis of *C. albicans* hyphae likely contributes to the stability of the barrier between CR3 and *C. albicans* β(1,3)-glucans, presumably by maintaining integrins in their active conformation (*Kaizuka et al., 2007*; *Lavi et al., 2007*; *Lavi et al., 2012*) during frustrated phagocytosis.

## Functional properties of the frustrated phagosome

Despite remaining unsealed, frustrated phagosomes acquired markers of endosomes and lysosomes, implying that they had undergone at least partial maturation. It was therefore conceivable that the cells established the diffusion barrier in an effort to generate a microbicidal compartment, despite their inability to form a sealed vacuole. Acidification of the lumen, secretion of antimicrobial enzymes and peptides and deployment of the NADPH oxidase are among the principal mechanisms used by leukocytes to eliminate pathogens. We first tested the ability of frustrated phagosomes to generate and maintain an acidic lumen, using the fluorescent acidotropic dye LysoBrite Red dye. As expected, the dye accumulated in lysosomes; however, it was never found to concentrate inside the frustrated phagosome (*Figure 10A*), suggesting that vacuolar ATPases are not functional on its membrane and/or that the junction separating the lumen from the extracellular milieu is permeable to $H^+$. That the latter interpretation is correct was suggested by determinations of permeability of the junction using dextrans of varying size. For these experiments lysosomes were loaded with either 10 kDa or 70 kDa fluorescent dextran and then exposed to *C. albicans* hyphae. The dextrans were delivered into fully formed (sealed) phagosomes, where they were clearly retained (*Figure 10—figure supplement 1A*). The smaller (10 kDa) dextran, however, was not detectable inside frustrated phagosomes; the reduced overall staining of the cells (*cf.* main panel and inset in *Figure 10B*) suggests that secretion of lysosomes did occur, but that the dextran must have escaped the confines of the frustrated phagosome. In contrast, the 70 kDa dextran was readily visible along the frustrated phagosome, implying that its diffusion into the external medium was limited. Thus, a size-selective filter determined the extent to which solutes were retained within the frustrated phagosome. The cut-off of this filter must be greater than ≈ 50 kDa, because cathepsin D, a globular protein of ≈ 28 kDa, managed to escape the frustrated phagosome (*Figure 10D*), yet was routinely detected in sealed phagosomes (*Figure 10—figure supplement 1B*). Therefore, the incomplete phagocytic cup formed around partially internalized hyphae would be expected to have limited degradative capacity towards *C. albicans*. These data are in accord with the findings of (*Prashar et al., 2013*) that showed frustrated *L. pneumophila* phagosomes to retain large molecular weight dextrans, but not protons or lysosomal enzymes, despite acquisition of the V-ATPase and fusion with lysosomes.

Though unable to retain luminal macromolecules over extended periods of time, the partial barrier to diffusion at the mouth of the frustrated phagosome, together with the geometrical constraint posed by the length and narrowness of the luminal space, are expected to delay the exit of molecules secreted into the phagosome. Rapidly reacting molecules may therefore be able to exert microbicidal/microbiostatic effects under these circumstances. Such is the case of reactive oxygen species produced by the NADPH oxidase. Indeed, we were able to detect preferential deposition of formazan, a product of the reaction of superoxide with nitroblue tetrazolium (NBT), inside frustrated phagosomes (*Figure 10E*). Heat-killed or paraformaldehyde-killed *C. albicans* hyphae were utilized for these experiments, eliminating the need to account for superoxide production by live *C. albicans* (see Materials and methods).

Because the frustrated phagosome appeared to retain some antimicrobial function, we assessed the effect of the frustrated phagosome environment on the fate of partially internalized hyphae. We

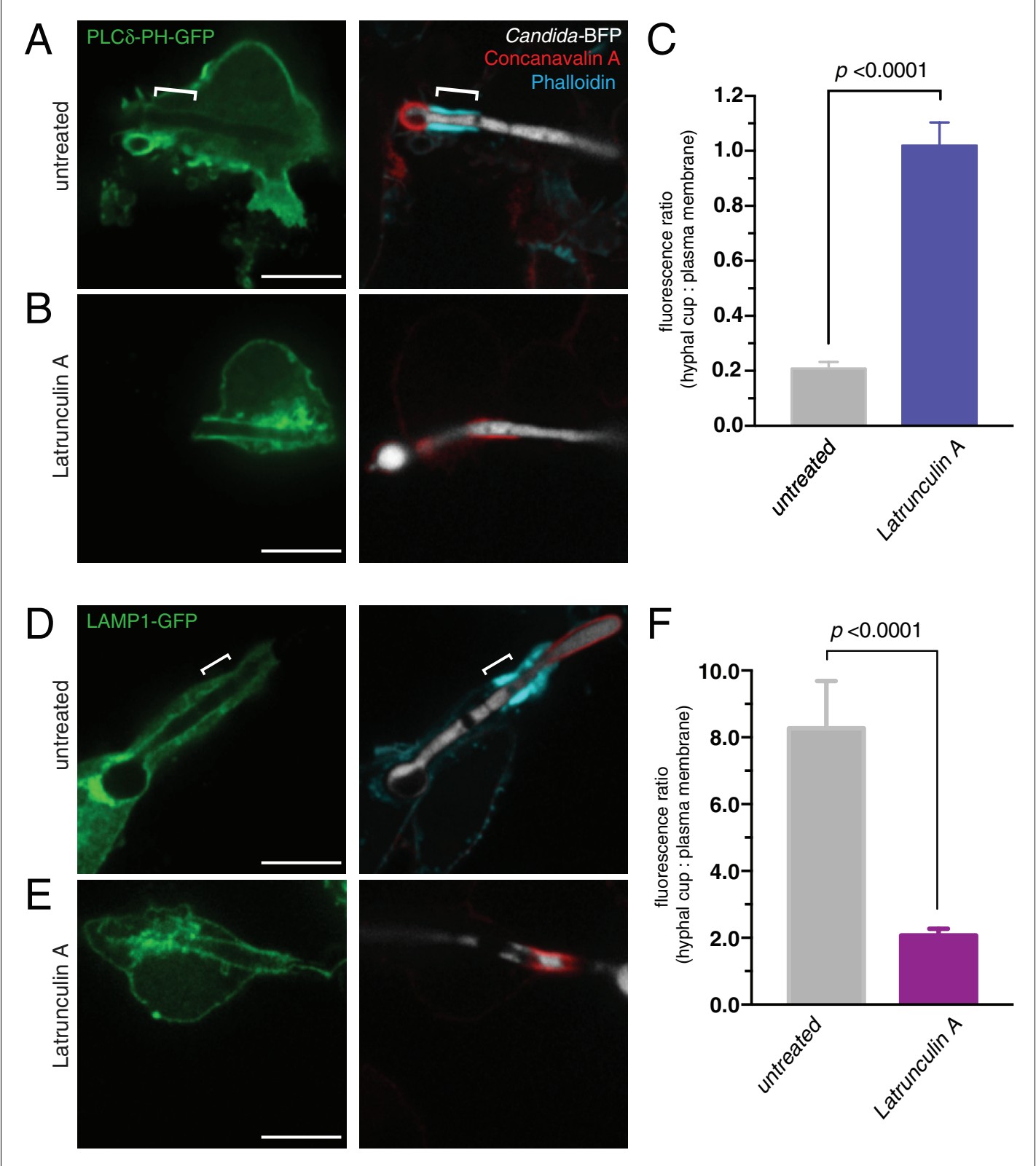

**Figure 9.** Actin depolymerization abolishes the diffusional barrier around *C. albicans* hyphae. RAW-Dectin1 cells were transfected with the indicated constructs, exposed to *Candida*-BFP hyphae and incubated for 30 min in the presence (**B and E**) or absence (**A and D**) of latrunculin A. After treatment, cells were fixed and extracellular *C. albicans* stained using Alexa594-conjugated concanavalin A (red), and actin stained using fluorescent phalloidin (blue). (**A and B**) Cells transfected with PLCδ-PH-GFP. (**C**) Effect of latrunculin A on actin cuff-mediated segregation of PLCδ-PH-GFP to the plasma

*Figure 9 continued on next page*

*Figure 9 continued*

membrane, quantitated as the ratio of the fluorescence intensity of GFP in the phagocytic cup over the plasma membrane. (**D and E**) Cells transfected LAMP1-GFP. (**F**) Effect of latrunculin A on actin cuff-mediated segregation of LAMP1 to the frustrated phagocytic cup, quantitated as the ratio of the fluorescence intensity in the phagocytic cup over the plasma membrane. For A and D, location of the actin cuff is indicated with a bracket. Scale bars: 10 μm. Images are representative of ≥30 fields from ≥3 separate experiments of each type. For each condition in C and F, three independent experiments were quantified, with ≥10 fields counted per replicate. *p* value was calculated using unpaired, 2-tailed students t-test. Data are means ±SEM.

DOI: https://doi.org/10.7554/eLife.34798.030

The following source data is available for figure 9:

**Source data 1.** Numerical data corresponding to *Figure 9C*.

DOI: https://doi.org/10.7554/eLife.34798.031

**Source data 2.** Numerical data corresponding to *Figure 9F*.

DOI: https://doi.org/10.7554/eLife.34798.032

could not detect significant loss of viability of the partially internalized *C. albican*s, as assessed by propidium iodide staining. We reasoned that the antimicrobial effectors may not suffice to kill the fungus, yet their effects may manifest as an observable change in the rate of hyphal extension, which can average 0.31 μm min$^{-1}$ on serum agar (*GOW and Gooday, 1982*). When measured in RPMI medium without serum (wthout macrophages present) *C. albican*s hyphae grew at a rate of 0.22 μm min$^{-1}$ ±0.03. Remarkably, the extension rate of partially internalized hyphae, which displayed an actin cuff, was significantly reduced (0.11 μm min$^{-1}$ ±0.01). This reduced growth rate was not different (p=0.742) to that of fully internalized hyphae (0.108 μm min$^{-1}$ ±0.011). In the same experiments, neighboring *C. albicans* hyphae not in contact with macrophages grew at a rate of 0.198 μm min$^{-1}$ ±0.014, indistinguishable from that measured in the absence of macrophages. Therefore, while small molecular weight contents can eventually diffuse out of the frustrated phagosome, they are nevertheless retained sufficiently to limit the growth of partially internalized *C. albicans*. This microbiostatic effect on partially internalized *C. albicans* hyphae could be ablated by blocking macrophage CR3 with the M1/70 antibody (see *Figure 3F,G and H*) before phagocytosis (*Figure 10—figure supplement 2*), reiterating the importance of CR3 ligation to β(1,3)-glucan for the generation and maintenance of this atypical phagocytic environment.

## Discussion

Most of the microbicidal and degradative properties of the phagosome depend on the release and containment of lysosomal hydrolases, antimicrobial peptides and reactive oxygen species in close proximity to the internalized microorganism. However, when phagocytes are faced with exceptionally large targets, their internalization can become retarded or frustrated altogether. The inability to complete phagocytosis, as in the case of long asbestos fibers (*Donaldson et al., 2010*) or bacterial biofilms (*Costerton et al., 1999*; *Scherr et al., 2014*; *Thurlow et al., 2011*) can potentiate harmful inflammation.

*C. albicans* hyphae can attain lengths of ≥50 μm (*GOW and Gooday, 1982*), overwhelming the comparatively diminutive phagocytes that are unable to ingest them whole. Accordingly, attempts to internalize such hyphae are frustrated and inflammatory in nature (*Branzk et al., 2014*; *Goodridge et al., 2011*; *Lewis et al., 2012*; *Rosas et al., 2008*). Nevertheless, our study shows that macrophages endeavor to seal the frustrated phagocytic compartment, in an effort to maximize their antimicrobial effect and minimize the release of inflammatory agents. To this end, they generate de novo a strikingly effective diffusion barrier by a process that involves activation of integrins that induce the formation of a thick F-actin cuff at the neck of the tubular phagosomes. The formation of actin-rich structures was reported previously during infection of macrophages with *C. albicans* (*García-Rodas et al., 2011*; *Heinsbroek et al., 2009*; *Strijbis et al., 2013*) and other rod/filament shaped microbes (*Gerisch et al., 2009*; *Prashar et al., 2013*), but neither their mechanism of assembly nor their functional significance were fully understood, which motivated our studies.

As expected, Dectin1 –the major phagocytic receptor for fungal β-glucan (*Brown and Gordon, 2001*; *Brown et al., 2002*; *Taylor et al., 2007*)– was present along the phagocytic cup, lining the internalized portion of the hyphae. Indeed, the RAW-Dectin1 cell line used for some of our studies

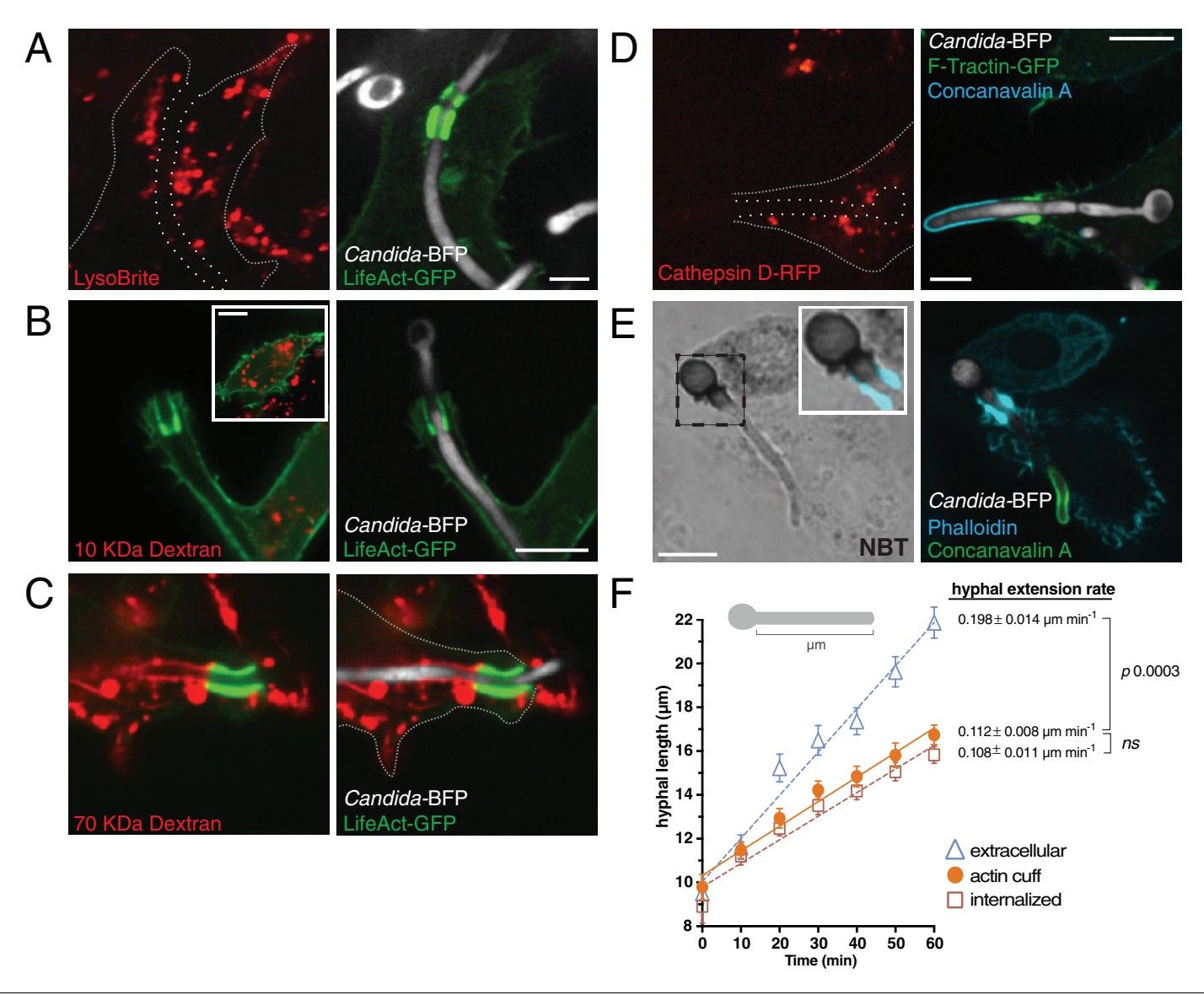

**Figure 10.** Analysis of the antimicrobial environment within the frustrated phagosome. (**A**) After phagocytosis of *Candida*-BFP hyphae by RAW-Dectin1 cells, acidic compartments were labeled with LysoBrite (red). The open hyphal phagocytic cup is marked with a dotted outline. Actin was visualized using transfected LifeAct-GFP. Scale bar: 5 µm. (**B and C**) Prior to phagocytosis of *C. albicans* hyphae, the lysosomes of RAW-Dectin1 cells were loaded with (**B**) 10 kDa or (**C**) 70 kDa fluorescent dextran (red). After phagocytosis, retention of dextran in the frustrated hyphal phagocytic cup was assessed by live cell microscopy. Actin was visualized using transfected LifeAct-GFP. Scale bars: 10 µm. (**D**) Retention of lysosomal hydrolases was assessed using transfected cathepsin D-RFP as a marker (red). Following phagocytosis and fixation, extracellular *C. albicans* was stained using Alexa647-conjugated concanavalin A (blue). Actin was visualized using transfected F-Tractin-GFP. Frustrated hyphal phagocytic cup marked with a dotted outline. Scale bar: 5 µm. (**E**) Generation of superoxide within the frustrated hyphal cup was detected using NBT. Following phagocytosis and fixation, extracellular *C. albicans* was stained using Alexa594 conjugated concanavalin A (red). Actin was stained using fluorescent phalloidin (blue). Inset shows merged image of formazan precipitate and the actin cuff. Scale bar: 5 µm. Images are representative of ≥30 fields from ≥3 separate experiments of each type. (**F**) Effect of the frustrated phagosome on *C. albicans* hyphal extension rate. After incubation with *Candida*-BFP hyphae for 10 min, RAW-Dectin1 cells transiently expressing F-Tractin-GFP were fixed at 10 min intervals, and extracellular *C. albicans* stained using fluorescent concanavalin A and visualized by confocal microscopy. The length of *C. albicans* hyphae (see (**F**), top) was measured for hyphae identified as extracellular (Δ), fully internalized (□) or partially internalized with actin cuffs (•), and the average hyphal length at each time-point and hyphal extension rate calculated. Average number of *C. albicans* per time-point was 55.0 ± 3.1. For each condition, four independent experiments were quantified, with ≥10 fields (37.5x) counted per replicate. *p* values calculated using the unpaired, 2-tailed students t-test. Data are means ±SEM.

DOI: https://doi.org/10.7554/eLife.34798.033

The following source data and figure supplements are available for figure 10:

*Figure 10 continued*

**Source data 1.** Numerical data corresponding to *Figure 10F*.
DOI: https://doi.org/10.7554/eLife.34798.036
**Figure supplement 1.** Fully internalized C. albicans hyphae contain typical lysosomal markers.
DOI: https://doi.org/10.7554/eLife.34798.034
**Figure supplement 2.** Engagement of integrin CR3 is necessary for the microbiostatic environment of the *C. albicans* frustrated phagosome.
DOI: https://doi.org/10.7554/eLife.34798.035

was created to allow efficient internalization of fungal zymosan (*Esteban et al., 2011*), and has been used to study *C. albicans*-macrophage interactions (*Strijbis et al., 2013*). Dectin1 signaling leads to robust production of reactive oxygen species by the NADPH oxidase in response to fungal ligands (*Brown et al., 2002*; *Goodridge et al., 2011*; *Underhill et al., 2005*), accounting for our observation that superoxide is detected within the frustrated phagocytic cup. Remarkably, however, Dectin1 did not concentrate in the region of the membrane adjacent to the actin cuff where the diffusion barrier was established. Instead, integrin $\alpha_M\beta_2$ (CR3, or CD11b/CD18) was found to accumulate in this region.

Engagement of CR3 at the cuff is required for the formation of the underlying actin cuff, a process likely mediated by talin and vinculin, which were also found accumulated at the site. The entire assembly appears central to the establishment of the diffusion barrier, which is lost when blocking CR3 binding with the M1/70 antibody and also when latrunculin is used to disassemble actin filaments. CR3 is unique amongst integrins in that its $\alpha$ domain contains a lectin-like domain (LLD) capable of binding fungal β-glucan (*Ross et al., 1985*; *Vetvicka et al., 1996*). This LLD, located at the membrane-proximal C terminus (between residues 943–1047; *Lu et al., 1998*) is distinct from the traditional ligand-binding I domain. Importantly, the LLD can bind β-glucan in a $Ca^{2+}$-independent manner while the integrin is in the inactive, bent conformation (*Thornton et al., 1996*). Binding of glucan has been shown to induce a semi-active conformation of the integrin that is predicted to facilitate outside-in signaling (*O'Brien et al., 2012*; *Vetvicka et al., 1996*). We found that Dectin1 did not have a direct role in actin cuff formation, being instead required for adhesion and the initiation of phagocytosis. Interestingly, when Dectin1 expression is low, the deposition of opsonins contained in serum –including complement and possibly also anti-*Candida* antibodies– suffice to engage CR3 and promote actin cuff formation directly activating this process, as predicted from earlier observations (*Boxx et al., 2010*; *Kozel et al., 1987*; *Vetvicka et al., 1996*). Dectin1 signaling can initiate inside-out activation of CR3 (*Li et al., 2011*). However, conventional Rap1-dependent inside-out signaling mediated by CalDAG-GEF1 was dispensable for actin cuff formation, as were divalent cations. Therefore, it is likely that CR3 binds fungal β-glucan in a manner that does not require inside-out activation by Dectin1.

We showed that mannan, a ligand for the mannose receptor, had no effect on actin cuff formation. The utilization of a curated set of *C. albicans* GRACE strains confirmed that mannan was dispensable and further excluded chitin and β(1,6)-glucan as ligands for cuff formation. Importantly, caspofungin inhibition of β(1,3)-glucan synthesis blocked formation of the cuff, in a dose dependent manner. These observations are in accord with involvement of the LLD of CR3, which was previously demonstrated to ligate β-glucan (*Mueller et al., 2000*; *Thornton et al., 1996*; *Vetvicka et al., 1996*). Clearly, β-glucan is also a ligand for Dectin 1 (*Brown and Gordon, 2001*; *Brown et al., 2002*; *Palma et al., 2006*), and initial engagement of this receptor is required for formation of actin cuffs around unopsonized hyphae. However, β(1,3)-glucan appears to play a distinct role in CR3-mediated actin cuff formation, as the effects of caspofungin could not be rescued by serum opsonization, with the caveat that caspofungin treatment may have affected complement deposition on *C. albicans* (*Boxx et al., 2010*; *Kozel et al., 1987*), although we regard this as unlikely because the fungal cell wall is rich in other polysaccharides and proteins that can serve to attach complement.

Our observations implicate clustered CR3 as an initiator of actin polymerization and a key constituent of the diffusion barrier. The signals mediating this effect include activation of Syk, which had been reported earlier (*Strijbis et al., 2013*), and also of Pyk2 and Fak. The latter related kinases were enriched at the cuff and dual inhibition by PF573228 blocked cuff formation. Along with activated Syk, Pyk2 and Fak have been shown to interact with β₂ integrins, including CR3 (*Duong and Rodan, 2000*; *Fernandez and Suchard, 1998*; *Han et al., 2010*; *Hildebrand et al., 1995*;

*Kamen et al., 2011*; *Mócsai et al., 2002*; *Raab et al., 2017*; *Rubel et al., 2002*; *Wang et al., 2010*; *Yan and Novak, 1999*). Pyk2, in particular, is required for paxillin and Vav1 activation during integrin engagement and CR3-mediated phagocytosis (*Kamen et al., 2011*). Paxillin was proposed to act as a scaffold, bridging integrin-initiated complexes with Rho-GTPases (*Deakin and Turner, 2008*), and Vav1, previously identified as important for the phagocytosis of *C. albicans* (*Strijbis et al., 2013*), also links $\beta_2$ integrins to the activation of Cdc42, Rac1 and RhoA (*Gakidis et al., 2004*). These signaling events appear conserved during the frustrated phagocytosis of *C. albicans* hyphae, as we detected paxillin and active Rac1/Cdc42 at the actin cuff, and Vav1 was found to be enriched in actin cuff-like structures around *C. albicans* (*Strijbis et al., 2013*).

The activation of the Rho family GTPases is linked to both formin- and Arp2/3- mediated actin dynamics. The actin cuffs formed around *C. albicans* hyphae were singularly sensitive to SM1-FH2 – and therefore dependent on formin-mediated linear actin polymerization– and not to inhibitors of the Arp2/3 complex that promotes branched actin polymerization. Interestingly, Rac1 and Cdc42 interact with actin-nucleating formins of the mDia family (*Lammers et al., 2008*). Collectively, our findings support a model whereby CR3 initiates signaling through Syk and Pyk2/Fak, leading to activation of Vav1 and Rho GTPases, culminating in formin-dependent actin nucleation.

Our studies show that the integrin/actin cuff separates the open phagocytic cup from the plasma membrane, appearing to act as a boundary that segregates distinct and mobile membrane domains. There was a clear segregation of phosphoinositides between the plasma membrane (PtdIns(4,5)$P_2$) and the open cup (PtdIns(3,4,5)$P_3$ and PtdIns(3)P). In principle, such separation could stem from the differential and strategic localization of kinases and/or phosphatases in the two membranes and in the junctional complex. However, we also observed slowly convertible (LC3) or non-convertible lipid-anchored proteins (constitutively-active Rab7) and transmembrane proteins (LAMP1) to be retained in the phagocytic cup, unable to reach the surface membrane. FRAP studies confirmed that these molecules were freely mobile within the phagocytic cup, pointing to the integrin/actin cuff as the diffusion barrier.

While antibody-induced clustering of CR3 was sufficient to form an actin-independent diffusional barrier, actin was required to maintain the barrier function of the cuff during phagocytosis of hyphae. Actin can stabilize integrins in their active conformation (*Kaizuka et al., 2007*; *Lavi et al., 2007*; *Lavi et al., 2012*), and such stabilization is likely required to maintain the cuff during extended frustrated phagocytosis. Actin-dependent diffusional barriers have been invoked in other systems (*Golebiewska et al., 2011*; *Nakada et al., 2003*; *Prashar et al., 2013*), although the pickets that anchor the cytoskeletal fence and restrict the diffusion of membrane components had not been previously identified.

We also analyzed the functional consequences of the establishment of the cuff and diffusion barrier. By segregating the two domains, the barrier enabled the open phagocytic cup to undergo an atypical maturation, despite the fact that scission from the surface membrane never occurred. This enabled targeting and activation within the frustrated phagosome of the NADPH oxidase, which generated toxic superoxide in the immediate vicinity of the portion of the hypha that had been engulfed. Thus, a crucial means for *C. albicans* control during infection (*Sasada and Johnston, 1980*; *Brothers et al., 2013*) remains operational in the frustrated phagosomes.

The observation that the tubular phagosomes were rich in LAMP1, a prototypic lysosomal marker, suggests that lysosomal hydrolases must have been secreted also into the phagosomal lumen. These were, however, not well retained by the phagosomes because, despite the tight seal that separated the inner leaflet of the membrane, the junction separating the aqueous compartments (i.e. the lumen from the extracellular space) was not perfectly tight. While 70 kDa dextran was retained within the phagosome, 10 kDa dextran was not, resembling the findings in frustrated *L. pneumophila* phagosomes (*Prashar et al., 2013*) and indicating the establishment of a sieve that excluded molecules with a hydrodynamic radius greater than ≈ 6–8 nm (*Nicholson and Tao, 1993*). Cathepsin family members (radius ≈ 2.4 nm; *Fazili and Qasim, 1986*) and similarly-sized hydrolases would therefore eventually escape the lumen. Nevertheless, because a partial seal is formed, their rate of loss might be slowed, allowing hydrolases and other antimicrobial molecules to act on the partially internalized *C. albicans* hyphae before exiting the open cup. Fast-acting antimicrobial agents, like ROS, released in close proximity to the target would be expected to be at least partially effective. Consistent with this hypothesis, partially internalized *C. albicans* hyphae exhibited a reduced growth rate compared to external hyphae. Importantly, this growth restriction was abolished upon antibody blockade of

CR3 and loss of actin cuff formation, presumably a result of increased leakage of phagosome contents. In this case, agents such as ROS, would not reach sufficient concentration to manifest the microbiostatic effect. However, leakage of phagosomal contents or ROS does not explain the failure of the frustrated phagosomes to kill the fungus, because *C. albicans* yeast and hyphae survive also within fully sealed phagosomes.

Based on the preceding considerations, we hypothesize that the integrin/actin cuff is generated and maintained by macrophages as a means to sustain antimicrobial functions in open tubular phagosomes formed around *C. albicans* hyphae and possibly other targets. It is tempting to speculate that the unique conditions established by the diffusion barrier might provide additional benefits to the phagocyte. In dendritic cells, decreased phagosomal proteolysis associated with reduced phagosome acidification protects antigens for enhanced presentation (*Mantegazza et al., 2008*; *Rybicka et al., 2012*; *Savina et al., 2006*). In addition, the frustrated yet maturing phagosome may enable activation of endomembrane Toll-like receptors (TLRs). TLR3 and TLR9 both localize to intracellular compartments and recognize *C. albicans* nucleic acids and chitin, respectively, contributing to a protective cytokine response to the fungus (*Nahum et al., 2011*; *Wagener et al., 2014*). Interestingly, Dectin1 can collaborate with plasmalemmal TLRs (TLR2 or TRL4) to enhance signaling and augment cytokine production (*Ferwerda et al., 2008*; *Netea et al., 2006*; *Underhill, 2007*) and a similar synergy may apply to endomembrane TLRs. Indeed, Dectin1 recognition is required for TLR9 localization to the *C. albicans* phagosome and TLR9-dependent gene expression (*Khan et al., 2016*). Thus, the unique structure described here may play an important role in the control of fungal infection and possibly also in the management of biofilms and other large targets by phagocytes.

# Materials and methods

## Reagents

Mammalian expression vectors were obtained from the following sources: Emerald-Dectin1 (plasmid #56291; Addgene, Cambridge, MA), PAK-PBD-YFP (*Srinivasan et al., 2003*), E-cadherin-GFP (plasmid #67937; Addgene), β-catenin-GFP(plasmid #16071; Addgene), Talin-GFP (*Franco et al., 2004*), AKT-PH-GFP (*Marshall et al., 2001*), PLCδ-PH-GFP (*Botelho et al., 2000*), PX-GFP (*Kanai et al., 2001*), LC3-GFP (*Kabeya et al., 2000*), Rab7-GFP (*Bucci et al., 2000*), Rab7(Q67L)-RFP (*D''Costa et al., 2015*), Lamp1-GFP (*Martinez et al., 2000*), Lyn$_{11}$-GFP (*Teruel et al., 1999*), cathepsin D-RFP (*Yuseff et al., 2011*), LifeAct-RFP or -GFP (*Riedl et al., 2008*), F-tractin-GFP (*Belin et al., 2014*), CD2-CD45-GFP (*Cordoba et al., 2013*).

Primary antibodies were purchased from the following vendors: HA (catalogue #MMS-101P; Covance, Princeton, NJ), pTyrosine (catalogue #05–321; EMD Millipore, Billerica, MA), pFAK-Y397 (catalogue #3283S; Cell Signaling, Beverly, MA), pPYK2-Y402 (catalogue #3291S; Cell Signaling), pSFK-Y418 (catalogue #44660G; Invitrogen, Carlsbad, CA), pSYK-Y525/526 (catalogue #2771S; Cell Signaling), Talin (catalogue #T3287; Sigma-Aldrich, St. Louis, MO), Vinculin (catalogue #MAB3574; EMD Millipore), HS1 (catalogue #4557S; Cell Signaling), LAMP1 (catalogue # 1D4B-s, Developmental Studies Hybridoma Bank, Iowa City, IA), actin (catalogue #A4700; Sigma-Aldrich), CD11b (catalogue #557394; BD Biosciences, Franklin Lakes, NJ), CD18 (catalogue #557437; BD Biosciences), rat IgG$_{2B}$ isotype control (catalogue #MAB0061; R and D systems, Minneapolis, MN), paxillin (catalogue #P13520; Transduction Laboratories, Lexington, KY), GAPDH (catalogue #MAB374; EMD Millipore), E-cadherin (catalogue #610181; BD Biosciences), β-catenin (catalogue #610153; BD Biosciences). Unconjugated and Alexa488, Cy3, Cy5, HRP-conjugated secondary antibodies against mouse, goat, rat, rabbit IgGs were obtained from Jackson ImmunoResearch Labs (West Grove, PA).

## Fungal strains and culture conditions

A list of all *C. albicans* strains tested is provided in *Table 1*. *C. albicans* strain SC5314 expressing BFP (*Candida*-BFP; *Strijbis et al., 2013*) was grown at 30°C in YPD (BD Biosciences). *C. albicans* cell wall mutant strains were obtained from the GRACE collection of tetracycline-repressible mutant strains (*O'Meara et al., 2015*; *Roemer et al., 2003*). Depletion of target gene expression was achieved by adding 0.5 µg mL$^{-1}$ doxycycline (DOX) to the growth medium. To induce hyphae of *C. albicans*, overnight cultures were subcultured 1:1000 in RPMI-1640 medium and incubated at 37°C for 1–3 hr, as indicated in the text. In some cases, caspofungin (Sigma-Aldrich) was used to

pharmacologically inhibit β(1,3)-glucan synthesis. *C. albicans* overnight cultures were subcultured into RPMI-1640 containing 10, 5, 2.5, 1.25 and 0 ng $mL^{-1}$ caspofungin for 2 hr at 30°C. Cultures were then moved to 37°C for 1 hr to induce hyphae in the presence of caspofungin. To measure the effect of caspofungin on β(1,3)-glucan levels, *C. albicans* hyphae-infected wells were stained with 0.05% aniline blue (*EVANS et al., 1984*; *Lee et al., 2016*) overnight.

*A. fumigatus* strain AF293 (clinical isolate) was grown on YPD agar (Bioshop, Burlington, ON) plates at 30°C. Conidia were harvested in PBS containing 0.01% Tween-80. For experiments, resuspended conidia were diluted 1:10 in RPMI-1640 containing 0.01% Tween-80, and allowed to form hyphae at 30°C overnight. Hyphae were then washed twice with PBS 0.01% Tween-80, and diluted 1:10 or 1:100 into RPMI-1640 containing 0.01% Tween-80.

## Mammalian cells and culture conditions

The RAW 264.7 cell line was obtained from and authenticated by the American Type Culture Collection (ATCC, Manassas, VA). The RAW-Dectin1-LPETG-3xHA cell line (RAW-Dectin1) was provided by Dr. Karin Strijbis and authenticated for Dectin1-HA expression and Dectin1-mediated phagocytic ability by flow cytometry (*Esteban et al., 2011*). Prior to experimentation, these cell lines were validated in our laboratory by assessing their morphology, phagocytic ability and expression of plasma membrane markers. RAW 264.7 and RAW-Dectin1 cells were grown in RPMI-1640 medium containing L-glutamine (MultiCell, Wisent, St. Bruno, QC) and 10% heat-inactivated fetal calf serum (FCS; MultiCell, Wisent), at 37°C under 5% $CO_2$. The A431 cell line was obtained from and authenticated by the American Type Culture Collection (ATCC). Prior to experimentation, this cell line was revalidated by assessing its expression of plasma membrane markers, and responsiveness to epidermal growth factor (EGF). A431 cells were grown in DMEM medium containing L-glutamine (MultiCell, Wisent) and 10% heat-inactivated FCS, at 37°C under 5% $CO_2$. All cell lines tested negative for mycoplasma contamination by DAPI staining.

Bone marrow-derived macrophages (BMDM) were obtained from the femoral bones of CALDAG-GEF1$^{-/-}$ (*Bergmeier et al., 2007*) or $^{+/+}$ (C57BL/6) mice, and differentiated for 5–7 days in DMEM containing L-glutamine, 10% heat-inactivated FCS, 100 U $mL^{-1}$ penicillin, 100 µg $mL^{-1}$ streptomycin, 250 ng $mL^{-1}$ amphotericin B (MultiCell, Wisent) and 10 ng $mL^{-1}$ mM-CSF (PeproTech, Rocky Hill, NJ), at 37°C and 5% $CO_2$. To obtain M2 human monocyte-derived macrophages, peripheral blood mononuclear cells were isolated from the blood of healthy donors by density-gradient separation with Lympholyte-H (Cedarlane, Burlington, ON). Human monocytes were then separated by adherence, and incubated in RPMI-1640 containing L-glutamine, 10% heat-inactivated FCS, 100 U $mL^{-1}$ penicillin, 100 µg $mL^{-1}$ streptomycin, 250 ng $mL^{-1}$ amphotericin B and 25 ng $mL^{-1}$ hM-CSF (PeproTech) for 7 days.

## Phagocytosis

Mammalian cell lines or primary cells were seeded on 18 mm coverslips in 12-well plates at $2 \times 10^5$ cells $mL^{-1}$. For infections with *C. albicans*, the medium was aspirated from the wells and replaced with 1 mL of *C. albicans* that had been induced to form hyphae. Plates were centrifuged for 1 min at 1500 rpm, then incubated with the following cell types at 37°C and 5% $CO_2$ for phagocytosis to proceed: RAW-Dectin1: 1 hr; BMDM: 15 min; human M2 macrophages: 20 min; A431 cells: 3 hr. In some cases, 30 min prior to infection, *C. albicans* hyphae were opsonized in human serum to promote the deposition of complement, although deposition of donor-specific anti-*Candida* antibodies may have also occurred, further favoring phagocytosis.

Where indicated, cells were treated with either vehicle or 1 µM Latrunculin A (Sigma-Aldrich) after 1 hr *Candida*-BFP infection, or pretreated 30 min with 4 mM EDTA (Bioshop), followed by infection with *Candida*-BFP in the presence of EDTA. For inhibition of actin polymerization or kinases, after 30 min incubation with *Candida*-BFP, monolayers were treated with either vehicle, 50 µM CK-666 (Calbiochem, La Jolla, CA), 10 µM SMI-FH2 (Calbiochem), 10 µM PP2 (Calbiochem), 50 µM piceatannol (Sigma Aldrich), or 50 µM PF573228 (Tocris, Oakville, ON), for 30 min. Following infection, monolayers were washed three times with PBS and fixed with 4% paraformaldehyde (PFA). In some cases, wells were treated with various fluorescent reagents before or after phagocytosis, as described below.

For infections with *A. fumigatus*, RAW-Dectin1 cells were incubated with 1 mL diluted *A. fumigatus* hyphae, and plates centrifuged for 5 min at 1500 rpm. Plates were incubated for 2 hr at 37°C and 5% $CO_2$ for phagocytosis to proceed. RAW-Dectin1 cells were pretreated 10 min and infected in the presence of 100 mM L-cysteine (Sigma-Aldrich) to prevent gliotoxin-mediated inhibition of phagocytosis (*Schlam et al., 2016*). Following infection, monolayers were washed three times with PBS and fixed with 4% paraformaldehyde (PFA). In some cases, wells were treated with various fluorescent reagents before or after phagocytosis, as described below.

After phagocytosis, external *C.albicans* were labeled for 20 min at room temperature using a solution of 5 µg mL$^{-1}$ fluorescent conjugated concanavalin A (ThermoFisher Scientific, Waltham, MA). To stain actin filaments, cells were permeabilized 5 min with 0.1% Triton X-100 and incubated 30 min with a 1:1000 dilution of fluorescent phalloidin (Thermofisher Scientific) or acti-stain (Cytoskeleton, Inc., Denver, CO). In some cases, *C. albicans* and *A. fumigatus* were stained with 10 µg mL$^{-1}$ calcofluor white (Fluorescent Brightener 28; Sigma-Aldrich).

## DNA transfection

For transient transfection, RAW-Dectin1 cells were plated on 18 mm glass coverslips at a concentration of $2 \times 10^5$ cells mL$^{-1}$ 16–24 hr prior to experiments. FuGENE HD (Promega, Madison, WI) transfection reagent was used according to the manufacturer's instructions. RAW-Dectin1 cells were transfected at a 3:1 ratio using 1.5 µL FuGENE HD and 0.5 µg DNA per well, and used for experiments 16 hr after transfection.

In some cases, DNA transfections were performed using the Neon transfection system (Life Technologies, Carlsband, CA) according to the manufacturer's protocol. RAW-Dectin1 cells were lifted, washed and resuspended to a concentration of $4 \times 10^6$ cells mL$^{-1}$ and 100 µL of the suspension were mixed with 5 µg DNA. Electroporation was done using a single 20 ms pulse of 1750 V. Cells were then immediately transferred to RPMI-1640 containing L-glutamine and 10% heat-inactivated FCS, before seeding on coverslips at concentration of $2 \times 10^5$ cells mL$^{-1}$. Cells were used for experiments 16 hr after electroporation.

## Immunofluorescence

After phagocytosis, fixation and concanavalin A staining (as indicated), monolayers were permeabilized in PBS containing 0.1% Triton X-100 for 5 min and blocked in PBS containing 5% skim milk and 0.1% Triton X-100 for 30 min at room temperature. Samples were incubated with primary antibodies for 30 min at room temperature. Primary antibody dilutions were: HA (1:1000), pTyrosine (1:100), pFAK-Y397 (1:100), pPYK2-Y402 (1:100), pSRC-Y418 (1:100), pSYK-Y525/526 (1:100), talin (1:500), vinculin (1:500), HS1 (1:250), LAMP1 (1:20), actin (1:100), E-cadherin (1:100), β-catenin (1:100), CD11b (1:100), CD18 (1:100), paxillin (1:100). After rinsing with PBS, samples were incubated 30 min at room temperature with Alexa488, Cy3 or Cy5-conjugated secondary antibodies at a 1:10,000 dilution. Where indicated, fluorescent phalloidin at a 1:1000 dilution was included with secondary antibodies. Samples were rinsed and viewed in PBS by confocal microscopy.

## Immunoblotting

Cells were grown in six well plates at a concentration of $4 \times 10^5$ cells per well. After infections, wells were lysed in Laemmli buffer (Bio-Rad, Mississauga, ON). Samples were run on a 7% SDS-PAGE gel for separation and the gel was transferred to a polyvinylidene difluoride (PVDF) membrane. Membrane was blocked in PBS containing 5% skim milk and 0.05% Tween-20 for 30 min at room temperature, followed by primary antibody staining for 1 hr at room temperature, in blocking buffer. Primary antibodies dilutions: E-cadherin (1:10,000), β-catenin (1:1000), GAPDH (1:20,000; loading control). After washing membrane in PBS containing 0.05% Tween-20, samples were incubated 30 min at room temperature with HRP-conjugated secondary antibodies at a 1:3000 dilution. Blots were visualized using the ECL Prime Western Blot detection reagent (GE Healthcare, Mississauga, ON) on an Odyssey Fc (LI-COR, Lincoln, NE).

## Rhodamine-PtdEth labeling

5 µg Rhodamine-PtdEth (L-α-phosphatidylethanolamine-N-(lissamine rhodamine B sulfonyl) ammonium salt; Avanti Polar Lipids, Inc., Alabaster, AL) was dried under $N_2$ and resuspended in 10 µL

methanol. After vortexing, 900 µL 3 mg mL$^{-1}$ bovine serum albumin (BSA) was added. This was then diluted 1:1 in cold serum-free RPMI-1640 containing 25 mM HEPES (HPMI; MultiCell, Wisent). Following phagocytosis, the medium was aspirated and replaced with 500 µL of the prepared Rhodamine-PtdEth/HPMI, and incubated at 4°C for 10 min. The adherent cells were rinsed three times with cold HPMI, heated to 37°C 10 min and imaged live.

### Cholera toxin B labeling

Following phagocytosis, cells were rinsed three times with cold HPMI. Cholera toxin subunit B, Alexa488 conjugate (Thermofisher Scientific) was added to a final concentration of 1 µg mL$^{-1}$ and cells incubated at 4°C for 10 min. The adherent cells were rinsed three times with cold HPMI, heated to 37°C 10 min and viewed live.

### LysoBrite red staining

Following phagocytosis, acidic intracellular compartments were stained using a 1:5000 dilution of the acidotropic LysoBrite Red dye (AAT Bioquest, Sunnyvale, CA) for 5 min at 37°C. Monolayers were rinsed three times with PBS, placed in HPMI, and imaged live.

### Dextran loading

RAW-Dectin1 cells were pulsed overnight with 20 µg mL$^{-1}$ Alexa647-conjugated 10 kDa dextran or 25 µg mL$^{-1}$ tetramethylrhodamine-conjugated 70 kDa dextran. Cells were washed and incubated with *Candida*-BFP, as described above, for 1 hr. Following phagocytosis, monolayers were rinsed three times with PBS, placed in HPMI, and viewed live.

### Nitroblue tetrazolium assay

This assay required PFA-inactivated *Candida*-BFP hyphae, as metabolically active *C.albicans* reduced nitroblue tetrazolium (NBT) to formazan, confounding the results. *Candida*-BFP hyphae were made as described above, followed by fixation in 8% PFA for 20 min. PFA-inactivated hyphae were rinsed and used to infect RAW-Dectin1 cells for 1 hr, in the presence of 0.5 mg mL$^{-1}$ NBT (Sigma-Aldrich). Following infection, cells were washed three times with PBS and fixed with 4% PFA. Formazan precipitate, created in response to superoxide anion produced by phagosomal NADPH oxidase, was visualized by bright field microscopy.

### Blocking experiments

To block CR3, adherent RAW-Dectin1 cells were incubated with 10 µg blocking antibody to CD11b (monoclonal M1/70) or rat IgG$_{2B}$ isotype control (MAB0061; R and D systems) for 30 min at room temperature. After warming to 37°C, cells were incubated with *Candida*-BFP hyphae as described above. Following phagocytosis, monolayers were rinsed and fixed in 3% PFA as mentioned above. After permeabilization and blocking, described above, blocking antibody was detected using a Alexa488-conjugated secondary antibody against rat IgG (1:10,000; Jackson ImmunoResearch Labs), and viewed by confocal microscopy.

For sugar blocking experiments, adherent RAW-Dectin1 cells were pretreated 30 min with 100 µg mL$^{-1}$ mannan (Sigma Aldrich) or laminarin (Sigma Aldrich), followed by infection with *C.albicans* in the continued presence of each sugar. In the case of laminarin, *Candida*-BFP hyphae were allowed to adhere to RAW-Dectin1 cells 10 min, followed by the addition of 100 µg mL$^{-1}$ laminarin for the remainder of the 1 hr infection.

### CR3 patching

RAW-Dectin1 cells transiently transfected with CD2-CD45-GFP were washed with PBS and cooled to 10°C in 1X HBSS (MultiCell, Wisent). To crosslink surface CR3, cells were sequentially incubated with: (1) 5 µg mL$^{-1}$ anti-CD18 primary (monoclonal M18/2); (2) goat anti-rat secondary and (3) donkey anti-goat tertiary antibodies, for 30 min each at 4°C, in PBS containing Ca$^{2+}$, Mg$^{2+}$ and 0.1% glucose. Labeled cells were then incubated 30 min at 37°C in the presence of 30 µM DYNGO 4a, to patch crosslinked CR3 without its internalization. Following this treatment, cells were treated 10 min with or without 1 µM latrunculin A, in the presence of 1 µM N-ethylmaleimide to prevent internalization of crosslinked CR3 following actin depolymerization. Monolayers were fixed in 3% PFA, and

extracellular CR3 was labeled using a fluorescent anti-goat antibody. Plasma membrane clusters of CR3 were analyzed in Volocity software for exclusion of CD2-CD45-GFP. Exclusion was calculated from confocal slices as the ratio of the GFP intensity per pixel in the CR3-positive patches to the average intensity in the plasma membrane.

## Hyphal growth rate of *C. albicans* during infection

RAW-Dectin1 cells were incubated with *Candida*-BFP hyphae for 10 min to allow adherence, then fixed in 4% PFA after 0, 10, 20, 30, 40, 50 or 60 min. Following phagocytosis, the cells were fixed in 3% PFA as above, external *Candida*-BFP were labeled with fluorescent concanavalin A, and permeabilized and stained with fluorescent phalloidin, as described above. Samples were imaged by confocal microscopy, and the length of individual *Candida*-BFP hypha measured in µm. Hyphal extension rate was calculated by linear regression analysis in GraphPad Prism software (GraphPad Software, Inc., La Jolla, CA).

## Confocal microscopy

Confocal images were acquired using a Yokogawa CSU10 spinning disk system (Quorum Technologies Inc., Guelph, ON) or a Leica SP8 laser scanning system (Leica). Images were acquired using a 63x/1.4 NA oil objectives or 25x/0.8 NA water objective (ZEISS, Germany), as indicated, with an additional 1.5x magnifying lens. For live experiments, cells were maintained at 37°C using an environmental chamber (Live Cell Instruments, Korea). Routine analyses and colocalizations were done using Volocity software (Perkin Elmer, Woodbridge, ON). 3D data visualization was done using Imaris (Bitplane, Concord, MA) software.

For colocalization analyses, Volocity software was used to calculate positive product of the differences of the mean (*Li et al., 2004*) channels, which were then overlayed on merged images for visualization.

For fluorescent intensity calculations, background subtracted intensities per unit area for expressed fluorescent protein constructs or endogenous proteins (immunofluorescence) were measured in Volocity software. Ratios were calculated comparing relative intensities in the actin cuff compared to phagocytic cup, or phagocytic cup compared to membrane, as indicated in the text.

## Fluorescence recovery after photobleaching (FRAP)

FRAP experiments of GFP or RFP-tagged proteins, transiently expressed in RAW-Dectin1 cells, were conducted on an A1R point-scanning confocal system (Nikon Instruments, Japan). For FRAP, *Candida*-BFP hyphae-infected cells were imaged in HPMI at 37°C. Images were acquired using a 60x/1.4 NA oil objective (Nikon), 1.2-AU pinhole, resonant scanning mode, and 16x line averaging. For a complete 2 min FRAP acquisition at 1.9 fps, after 5 s of initial imaging, a region of interest 3 µm in diameter was bleached for 1.06 s using the 405 laser at 100% power, followed by imaging for fluorescence recovery. Images were exported and analyzed for fluorescence intensity using Volocity software.

After background subtraction, fluorescence intensity units were normalized (see *Figure 6* legend) using Microsoft Excel software, and transformed to a 0–1 scale, to correct for differences in bleaching depth and allow for comparison of up to 30 individual FRAP curves per condition. Graphpad Prism software was used to fit the FRAP curves to a single exponential, plotted as fractional recovery over time.

## Transmission electron microscopy (TEM)

*Candida*-BFP hyphae-infected RAW-Dectin1 cells were washed with cold PBS, and fixed with 2% glutaraldehyde in 0.1 M sodium cacodylate buffer, pH 7.3. For improved specimen preparation, samples were then subjected to a zymolyase digestion protocol (*Bauer et al., 2001*) designed to weaken the fungal cell wall and allow for adequate structural preservation. Samples were then postfixed in 1% osmium tetroxide in 0.1 M sodium cacodylate buffer, pH 7.3, dehydrated in a graded ethanol series followed by propylene oxide, and embedded in Quetol-Spurr resin. Ninety nm sections were cut on a Leica Ultracut ultramicrotome and stained with uranyl acetate and lead citrate. Samples were imaged on a FEI Tecnai 20 transmission electron microscope, equipped with an AMT 16000 digital camera.

## Acknowledgements

We sincerely thank Dr. Karin Strijbis (Utrecht University, The Netherlands), for providing RAW-Dectin1-LPETG-3xHA cells and *C. albicans* strain SC5314 expressing BFP, and Dr. Wolfgang Bergmeier (University of North Carolina), for providing tissue from CalDAG-GEF1 knockout mice. We thank Merck and Genome Canada for making the *C. albicans* GRACE collections available. MEM was the recipient of a Heart and Stroke Pfizer Research Fellowship. TRO is supported by a National Institutes of Health (NIH) Ruth L Kirschstein National Research Service Award (NRSA, AI115947-01) from the NIAID. LEC is supported by Canadian Institutes of Health Research Operating Grants (PJT-153403, PJT-148548, MOP-86452, and MOP-119520) and Foundation Grant (FDN-154288), the Natural Sciences and Engineering Council (NSERC) of Canada Discovery Grants (06261 and 462167), an NSERC EWR Steacie Memorial Fellowship (477598), and a National Institutes of Health NIAID R01 (1R01AI127375-01). SG is supported by Canadian Institutes of Health Research grant FDN–143202.

## Additional information

### Funding

| Funder | Grant reference number | Author |
|---|---|---|
| Canadian Institutes of Health Research | FDN143202 | Sergio Grinstein |
| Heart and Stroke Foundation of Canada | Heart and Stroke Pfizer Fellowship | Michelle E Maxson |
| National Institutes of Health | AI115947-01 | Teresa R O'Meara |
| The Research Training Group 1459 | | Xenia Naj |
| Natural Sciences and Engineering Research Council of Canada | 06261 | Leah E Cowen |
| Canadian Institutes of Health Research | FDN-154288 | Leah E Cowen |
| Canadian Institutes of Health Research | PJT-153403 | Leah E Cowen |
| National Institutes of Health | 1R01AI127375-01 | Leah E Cowen |
| Natural Sciences and Engineering Research Council of Canada | 477598 | Leah E Cowen |
| Natural Sciences and Engineering Research Council of Canada | 462167 | Leah E Cowen |
| Canadian Institutes of Health Research | PJT-148548 | Leah E Cowen |
| Canadian Institutes of Health Research | MOP-86452 | Leah E Cowen |
| Canadian Institutes of Health Research | MOP-119520 | Leah E Cowen |

The funders had no role in study design, data collection and interpretation, or the decision to submit the work for publication.

### Author contributions

Michelle E Maxson, Conceptualization, Formal analysis, Funding acquisition, Investigation, Visualization, Methodology, Writing—original draft, Project administration, Writing—review and editing; Xenia Naj, Formal analysis, Funding acquisition, Investigation, Writing—review and editing; Teresa R O'Meara, Conceptualization, Funding acquisition, Investigation, Methodology, Writing—review and editing; Jonathan D Plumb, Conceptualization, Formal analysis, Validation, Investigation,

Visualization, Writing—review and editing; Leah E Cowen, Conceptualization, Resources, Funding acquisition, Methodology, Writing—review and editing; Sergio Grinstein, Conceptualization, Formal analysis, Supervision, Funding acquisition, Methodology, Writing—original draft, Project administration, Writing—review and editing

### Author ORCIDs
Michelle E Maxson (iD) http://orcid.org/0000-0002-1493-490X
Sergio Grinstein (iD) http://orcid.org/0000-0002-0795-4160

### Decision letter and Author response
Decision letter https://doi.org/10.7554/eLife.34798.041
Author response https://doi.org/10.7554/eLife.34798.042

## Additional files

### Supplementary files
• Transparent reporting form
DOI: https://doi.org/10.7554/eLife.34798.037

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
