## [Decision Letter]

[Editors’ note: a previous version of this study was rejected after peer review, but the authors submitted for reconsideration. The first decision letter after peer review is shown below.]

Thank you for submitting your work entitled "An integrin-based diffusion barrier separates membrane domains enabling formation of microbicidal frustrated phagosomes" for consideration by *eLife*. Your article has been reviewed by three peer reviewers, and the evaluation has been overseen by a Reviewing Editor and a Senior Editor. The following individuals involved in review of your submission have agreed to reveal their identity: Neil Gow (Reviewer #1).

Our decision has been reached after consultation between the reviewers. Based on these discussions and the individual reviews below, we regret to inform you that your work will not be considered further for publication in *eLife*.

As you will see from their individual comments below, the reviewers concur that this study reports some advances the field of host-fungus interactions and add supporting data on role for integrin in the establishment or maintenance of the actin cuff. The manuscript is clearly written, and the work is well executed and clearly illustrated with beautiful imaging data.

However, the originality of the manuscript is in part compromised because some of the concepts were preceded by a study from Dr. Terebiznik's lab. While the current work needs to be more cross-referral to this, there is still novelty in the current submission leaning on the identification of Mac-1-dependent diffusion barriers formed between the hyphae cell wall and the plasma membrane of the macrophage. The experimental data supporting this model requires important revisions in order for it to be convincing (more quantitative data). Furthermore, data in support of the microbicidal properties of these frustrated phagosomes is missing. The roles of Dectin-1 and Integrin receptors in the formation of actin cuffs and barriers should investigated in more detail whereas it is optional to expand the study into inflammatory response.

While we are rejecting the paper as a result of the need for a substantial revision (longer than 2 months), we are broadly supportive of this manuscript and would consider a newly submitted form of this paper that would be treated as a revised manuscript.

Reviewer #1:

This paper addresses a well appreciated but univestigated conundrum. How do immune phagocytes interact with target cells that are much larger than they can envelop? The authors show that macrophages that are challenged with elongated hyphal cells of the fungus *Candida albicans* wrap around the end of a hypha and form an actin cuff that partially seals the phagocytic cup at the mouth of the tubular phagosome. This structure retains some of the ability to establish an antimicrobial environment within the unsealed tubular phagosome – for example allowing NAPPH oxidase activation, but not phagosome acidification. The actin cuff is shown to represent a diffusion barrier that segregates phosphoinositides between the phagocytes cell membrane and the open cup. Some elegant experiments investigate how this is segregation is achieved and regulated. Overall this is an interesting, novel and well executed study that clearly advances the field of host-fungus interactions.

I have a number of questions, suggestions and presentational issues.

1) The fungus studied is *Candida albicans* but is most commonly referred to as Candida. It would not be simply pedantic to request that it be referred to as *C. albicans* throughout since most Candida species cannot form elongated hyphae.

2) It would be interesting to know if similar actin cuff seals are formed when phagocytes encounter hyphae of Aspergillus fumigatus (interesting because this fungus has quite marked differences in cell wall composition).

3) The hyphae used are not very long (1 h or 2 h hyphae were used) and the macrophages shown seem to be invaginating one end. Macrophages will also wrap around the trunk of a very large hypha. Are actin cuffs formed here- and what is their architecture? It would be useful to include some images of longer hyphae of 4-6 hours incubation in RPMI.

4) Various *C. albicans* mutants are available with alterations in the cell wall that would help determine what surface components of the fungus are important for inducing the cuffs.

5) In many of the figures it is not possible to clearly evaluate the shape and position of the phagocyte relative to the fungus. It would be very helpful to show a DIC or phase image of the same interacting cells in order to really see the orientation of the hypha and phagocyte. I have no doubt that the descriptions in the text are accurate, but I have to somewhat take it on trust since I cannot see the outline of the two interacting cells clearly in all cases. (For example see Figure 3B,C; Figure 4B,E,F; Figure 5B,C,F; Figure 6E; Figure 8A,D and others).

6) The authors show that the CTL Dectin-1 is not present at the actin cuff, but phosphotyrosine accumulated at this site suggesting that some other pattern recognition receptor may drive actin assembly. It would be useful to verify that glucan phosphate or laminarin did not block actin cuff formation.

7) Since the mannose receptor is likely to engage with hypha mannan, (which is located in the outer cell wall) it would be interesting to know if this is located at this site.

8) I'm not sure that the TEM shown in Figure 1C can be seen as evidence of showing the actin cuff unless the actin is perhaps stained using colloidal gold.

9) My88-/- cells would be useful in assessing the possible involvement of TLRs.

10) The actin cuffs appear to be elongated in some images and much shorter or almost like spots in others. Do they retain their shape over time or contract to a spot?

Reviewer #2:

In this manuscript M Maxson and collaborators have characterized the phagocytosis of *Candida albicans* hyphae, which for long hyphae, cannot progress beyond the stage of phagocytic cup, leading to the formation of a frustrated phagosome. The manuscript is clearly written, and the findings illustrated with beautiful imaging data. The authors propose that actin cuffs in the frustrated phagosomes, which have been described in previous reports on the phagocytosis in *Candida albicans* (Bain et al., 2014 and Lewis et al., 2012), may act as diffusion barriers and contribute to membrane fences that allow the accumulation of ROS in the frustrated phagosomes. They also propose that these actin cuffs are the product of the atypical activation of MAC1receptors in the macrophages by ligands expressed on the cell wall of Candida and that MAC1 may restrict the free diffusion of lipids and membrane proteins along the membrane. The proposed model is sound, and it may explain possible microbicidal actions phagocytes may attempt when pathogens cannot be completely engulfed. However, there are many important points in the manuscript that require clarification and revision to better support the authors' model.

The authors claim that the frustrated phagosomes are microbicidal. Yet, there is no data on fungal survival to support this conclusion. Are the macrophage-trapped hyphae growing longer or dying as time passes?

The contribution of Candida hyphae to the formation of the frustrated phagosomes and actin cuffs must be assessed. Changes in Candida cell wall composition have been previously reported to affect macrophage actin dynamics (Bain et al., 2014). However, the authors are not considering or discarding a role for the yeast in the proposed model.

For the imaging data, even though the regions of interest are nicely indicated with brackets, since different markers don't always localize to same regions along the frustrated phagosome, having additional panels with channels merged would make it easier to visualize what the authors are illustrating.

In subsection “Signals driving actin cuff formation”, referring to Figure 2AB, the authors indicate, a lack of Dectin-1 at the actin cuffs. However Emerald-Dectin-1 can be seen in the cuff in Figure 2B, which is in good agreement with Strijbis et al., (2013) that also showed that Dectin-1 is enriched in the cuff region of the phagocytic cup, as well as in areas of enrichment outside the cuff region. To support their claim, the authors must present quantitative data on Dectin-1 distribution along the frustrated phagosome. Also, it is not indicated, but the imaging shown in Figure 2, and other figures, seems to be a single confocal z-plane. It may also be appropriate to include extended focus projections, considering the size of the hyphae, which could be included as supplementary files.

Clarifying the distribution of Dectin1 in the frustrated phagosomes, is critical because it affects the interpretation of data in Figure 2D and other results in the manuscript.

Figure 2D showing PAK-PBD and tyrosine phosphorylation in the actin cuffs are likely expected results since Strijbis et al., 2013, have shown that the GEF for CDC42 and RAC1, VAV1, as well as the tyrosine kinases Syk and BTK, are all recruited to cuffs. Quantitative data for the recruitment of these molecules to the frustrated phagosomes and data proving that their activities are involved in the formation or sustaining the cuffs will certainly help the proposed model.

Figure 3D shows that on average, for a given field, 6 hyphae are observed in frustrated phagosomes with actin cuffs and this is consistent throughout the manuscript. However, it is not clear how many individual events were analyzed. How frequently are these frustrated phagosomes observed? Do all hyphae engaged by macrophages have these structures associated with them?

Figure 4. For accumulation of receptors provide quantitative data. According to the authors, MAC1 activation signals for the formation of the actin cuffs. However, in Figure 4A, CD11b did not localize to the thicker section of the cuffs. Please clarify and indicate the frequency of this phenotype.

Figure 4G shows a shorter cup in the presence of CD11b blocking antibodies. The treatment could cause an inhibition of phagocytosis. Authors must demonstrate that phagocytosis is not affected by the treatment.

Figure 4G/H In cells treated with the M1/70 antibody, actin is still present in the frustrated phagosome. Is this a cuff? In fact, Concavalin A labeling is limited to the phalloidin boundary. This suggests the presence of a cuff/ barrier, even after CD11b treatment.

Is dectin1 distribution along the frustrated phagosome affected after CD11b is blocked? Blocking CD11b in dectin1 knockout cells will help address whether dectin1 contributes to this barrier.

Figure 5, subsection “Phospholipid segregation between the plasma membrane and the cuff-delimited phagosomal cup” states "Astonishingly, while PtdIns(4,5)P2 was present as expected in the surface membrane facing the extracellular milieu, it was undetectable in the invaginated section that constituted the frustrated phagosome (Figure 5A)". This and other characteristics of the cuff and the frustrated phagosomes reported in Figure 5 and Figure 8 are similar to those reported by Prashar et al., (2013). This must be acknowledged in the manuscript. It must be acknowledged that PtdIns(3,4,5)P3 distribution in the frustrated phagosome has been previously described by Strijbis et al. In Figure 5D, how do the authors explain the presence of LC3 in the actin cuff?

Subsection “The actin cuff forms a diffusional barrier to the movement of proteins and lipids” "The sharp boundary between the PtdIns(4,5)P2-rich surface membrane and the tubular membrane endowed with PtdIns(3,4,5)P3 and PtdIns(Bauer et al., 2001)P coincided with the location of the actin cuff, suggesting that the latter may function as a diffusion barrier".

Subsection “The actin cuff forms a diffusional barrier to the movement of proteins and lipids” Similarly, both wild-type Rab7 (Figure 5E) and constitutively-active Rab7 (not illustrated) are confined to the frustrated phagosomal tube, as was LAMP1 (Figure 5F)……..we considered it more likely that restricted diffusion accounted for the observations"

Differently to what the authors describe in the paragraph from above, the phagosomal markers are penetrating the barrier as in the case of Figure 5B and E and surpassing the cuffs in Figure 5C, E and F. As indicated before for other results the authors must present quantitative data on the distribution of the markers and extended focus projections to support the proposed model.

Figure 7 strongly supports authors' claim However, if latranculin A treatment allowed Lamp-1 and PtdIns(4,5)P2 to migrate to previously excluded zones, why concanavalin A is not following and labels the full extension of the hyphae? That will be the expected if the diffusion barriers in the cuffs are dismantled. Quantitative data is missing, and full cell imaging would be preferable to better illustrate the authors' claim.

Figure 8 The characteristics of the frustrated phagosome are remarkably similar to long cups described by Prasher et al., 2013 and Strijbis et al., 2003. This must be cited in the text. The accumulation of NBT in Figure 8E is puzzling since small molecules can diffuse out across the actin cuff barrier. The authors must address the possibility that the NBT is accumulating inside the hyphae. NBT can accumulate in fungi Camile et al., 2008. Thus, authors must prove a link between ROS and microbicidal conditions in the long open phagosomes.

Reviewer #3:

The manuscript by Maxson et al., describes and analyzes partially sealed phagocytic cups in macrophages which form around *Candida albicans* hyphae. It demonstrates that an actin-rich cuff at the distal margin of the cup supports a lateral segregation between the components of the plasma membrane and the contiguous inner leaflet of the phagocytic cup. This lateral segregation excludes the PI(4,5)P2 from the cup and confines the PI(Bauer et al., 2001)P and PIP3 to the cup. The actin cuff contains the integrin CR3 and associated integrin signaling molecules. The barrier effectively limits the movement of inner leaflet probe molecules into or out of the cup, without significantly diminishing diffusion within the plasma membrane or the cup. The actin cuff can be disrupted by the actin depolymerizing drug latrunculin A. The barrier limits escape by diffusion of large macromolecules delivered into the lumen of the cup, but not the diffusive loss of smaller molecules, including dextrans and protons (pH). Nonetheless, reactive oxygen species can be delivered into the unclosed cups. The experimental work is carefully done and the morphological evidence in support of the claims is beautiful. However, much if not most of the conclusions reported here were first described in a large study by Prashar et al., (2013), using an analogous experimental model. Examining the interactions between macrophages and filamentous bacteria, that study demonstrated (Astarie-Dequeker et al., 1999) the actin cuff (called a "jacket") that segregates two domains of contiguous membrane (plasma membrane and phagocytic cup), (Bain et al., 2014) the effect of actin depolymerization on the maintenance of that segregation, (Bauer et al., 2001) the nature of the diffusion barrier with respect to extracellular probes and (Belin et al., 2014) the retention of diffusible molecules in the cup lumen (different sizes of dextrans, pH, hydrolytic enzymes). Moreover, unlike the present manuscript which shows that ROS can be generated in the unclosed cups, the previous paper measured the effect of partial phagocytosis on microbicidal activities against filamentous bacteria (albeit somewhat incompletely). The present manuscript but does not measure microbicidal activity against *C. albicans*.

Assuming this manuscript can be rewritten to acknowledge appropriately the demonstrated precedents and concepts of that earlier work, the present study does add some interesting new data which supports a role for integrin in the establishment or maintenance of the actin cuff. The role of integrin in the maintenance of the barrier remains indirect, however. Perhaps the actin ring is the main ingredient of the barrier, and anything which organizes actin into a ring can support the formation of diffusion barriers such as described here, and elsewhere for analogous structures (Golebiewska et al., 2011; Welliver et al., 2011. To better define the nature of the diffusion barrier, the diffusion measurements (e.g. Figure 6) or probe localization studies (e.g. Figure 5) described in the manuscript should be extended to analyze the roles of actin (latrunculin A) and integrin (M1/70) to barrier maintenance. This could provide a mechanistic underpinning to the diffusion measurements.

[Editors’ note: what now follows is the decision letter after the authors submitted for further consideration.]

Thank you for resubmitting your work entitled "Integrin-based diffusion barrier separates membrane domains enabling formation of microbiostatic frustrated phagosomes" for further consideration at *eLife*. Your revised article has been favorably evaluated by Ivan Dikic (Senior editor), a Reviewing editor, and three reviewers.

This revised manuscript contains extensive additional experimental work and text that address concerns noted in earlier reviews. The new work is thorough and adequately supportive of the conclusions. The manuscript now establishes a diffusion barrier in the inner leaflet of incompletely closed phagocytic cup membranes, comprised of CR3 integrins and maintained by an actin-rich cuff. Further, it identifies signaling molecules that contribute to cuff formation, and establishes the incompletely closed cups as microbiostatic. Related studies are adequately cited. The discussion provides a thoughtful analysis of the biology and its implications. Yet there are some minor remaining issues that need to be addressed in the text before acceptance, as outlined below:

The authors addressed all the critiques from the reviewers.

They reported additional results on the phagocytosis of *C. albicans* hyphae, improved the quality of the imaging, and provided numerical and statistical data to support their most relevant experiments.

However, below there are several points that we believe could be discussed in the manuscript:

1) In the abstract it is stated “[…]in response to non-canonical activation of integrins by fungal β(1,3)-glucans." We suggest using the singular "glycan". There is more than one form of fungal β(1,3)- glucan, but this is not relevant to this article.

2) B-glucans are involved in the activation of complement and the deposition of C3b fragments on the surface of *C. albicans*. (Boxx et al., 2010., Kozel et al., 1987, Vetvicka et al., 1996). What could be the contribution of C3b opsonization on the formation of the actin cuffs and the properties of the membrane and luminal diffusion barriers?

3) Previous reports on the phagocytosis of *C. albicans* showed the formation of PIPs in phagosomes containing this yeast (Heinsbroek, 2009). This was attributed to pathogenic mechanisms. On the other hand, and coinciding with the authors' interpretation, Naufer et al., 2018 recently reported that PI(Bauer et al., 2001)P co-exists with phagolysosomal markers in the phagocytic cup of heat-killed filamentous bacteria. Maybe the authors could mention this in their Discussion section.

4) Please clarify: In Figure 4, human serum was used to complement opsonize hyphae. How did the authors account for the presence of antibodies against C. albincans in the serum? Antibodies could be expected as this yeast is part of the human microbiome

5) Caspofungin treatment could prevent the deposition of complement in hyphae, and its recognition by antibodies and Dectin-1. This could explain the results in Figure 4—figure supplement 1.

6) The authors showed that ROS production in the phagocytic cup retarded the elongation of the hyphae. This was alleviated by blocking CR3 with monoclonal antibodies to impede the formation of cuffs, a procedure that favours a ROS leaching. Since ROS are generally considered microbicidal, perhaps, ROS failing to reach lethal concentrations in the open cup could be the cause of the effect reported by the authors. Therefore, assuming a microbiostatic mechanism is probably incorrect.

---

## [Author Response]

[Editors’ note: the author responses to the first round of peer review follow.]

Reviewer #1:I have a number of questions, suggestions and presentational issues.1) The fungus studied is Candida albicans but is most commonly referred to as Candida. It would not be simply pedantic to request that it be referred to as C. albicans throughout since most Candida species cannot form elongated hyphae.

As requested, we have changed our notation to *C. albicans* throughout the text, and only refer to the BFP-expressing *C. albicans* as *Candida*-BFP to preserve the original nomenclature coined by the authors that generated this strain (Strijbis, 2013).

2) It would be interesting to know if similar actin cuff seals are formed when phagocytes encounter hyphae of Aspergillus fumigatus (interesting because this fungus has quite marked differences in cell wall composition).

As suggested by the reviewer, we performed new experiments in collaboration with the laboratory of Dr. Leah Cowen (who is now included as a co-author) using *Aspergillus fumigatus*. The new data are shown in the revised Figure 4G and discussed in the Results section. Like others before, we observed that *A. fumigatus* is not effectively internalized by macrophages, due in part to the presence of cell wall glycosaminoglycans and also because it produces gliotoxin, an effective inhibitor of phagocytosis. We minimized the effects of gliotoxin (that reacts with sulfhydryl groups, forming mixed disulfides) by exposing the cells to *Aspergillus fumigatus* in the presence of L-cysteine (see Materials and methods section). Under these conditions, when the macrophages engage the hyphae near the end, we were able to observe the formation of distinct actin cuffs, resembling those formed around *C. albicans* hyphae, consistent with the fact that the cell wall of *A. fumigatus* contains β(1,3)-glucans.

3) The hyphae used are not very long (1 h or 2 h hyphae were used) and the macrophages shown seem to be invaginating one end. Macrophages will also wrap around the trunk of a very large hypha. Are actin cuffs formed here- and what is their architecture? It would be useful to include some images of longer hyphae of 4-6 hours incubation in RPMI.

Based on this suggestion, we conducted experiments using longer *C. albicans* hyphae (allowed to grow for 4 hours), and also observed the formation of actin cuffs when these were engaged near the end. A representative image is shown in Author response image 1 (top row). In the case of the longer hyphae, as was also the case with the very long *Aspergillus* hyphae, the macrophages often interacted with intermediate regions of the hyphae. In these instances, the macrophages partially wrapped themselves around the hyphae, forming dense actin structures that did not seem to surround the hyphae in their entirety (we call them “tacos”; see Author response image 1, bottom row, for representative image). Because these seemingly unsealed structures would not function as diffusion barriers –the main topic of the paper– we have opted not to illustrate or discuss them but would be happy to do so if the reviewer felt this is important.

4) Various C. albicans mutants are available with alterations in the cell wall that would help determine what surface components of the fungus are important for inducing the cuffs.

As recommended by the reviewer we have conducted an extensive series of experiments in collaboration with Drs. L. Cowen and T. O’Meara to define the *C. albicans* cell wall components required for actin cuff formation. To this end we utilized a curated set of *C. albicans* GRACE strain mutants generously made available by Merck and Genome Canada, which filament normally but have mutations affecting biosynthetic pathways for chitin, mannans or β(1,6)glucans (see O’Meara et al., 2015). The new results are now illustrated in Figure 4C and D and described in the Results section. To assess the role of β(1,3)-glucans, we pharmacologically inhibited β(1,3)-glucan synthesis using caspofungin. These results are presented in the new Figures 4E and F and discussed in the Results section. In a nutshell, we found that β(1,3)-glucan is required for actin cuff formation, while chitin, mannans and β(1,6)-glucans were dispensable. Please note that we also utilized caspofungin in Figure 4—Figure supplement 1, discussed in the Results section, to distinguish the role of β(1,3)-glucan in actin cuff formation from its role in Dectin1-mediated phagocytosis.

5) In many of the figures it is not possible to clearly evaluate the shape and position of the phagocyte relative to the fungus. It would be very helpful to show a DIC or phase image of the same interacting cells in order to really see the orientation of the hypha and phagocyte. I have no doubt that the descriptions in the text are accurate, but I have to somewhat take it on trust since I cannot see the outline of the two interacting cells clearly in all cases. (For example see Figure 3B,C; Figure 4B,E,F; Figure 5B,C,F; Figure 6E; Figure 8A,D and others).

As requested by the reviewer, to enable the reader to locate the phagocytes with respect to the hyphae, we have outlined the macrophages using grey dotted lines (see Figure 1, Figure 1—figure supplement 1, Figure 2, Figure 3, Figure 3—figure supplement 1, Figure 4, Figure 5, Figure 5—figure supplement 1, Figure 6 and Figure 10), except in those cases where the cell outline is readily discernible as background fluorescence.

6) The authors show that the CTL Dectin-1 is not present at the actin cuff, but phosphotyrosine accumulated at this site suggesting that some other pattern recognition receptor may drive actin assembly. It would be useful to verify that glucan phosphate or laminarin did not block actin cuff formation.

Considering this comment, together with comment #7 below, we performed additional experiments to characterize the properties of the receptor-ligand interaction involved. As suggested, we used laminarin to block Dectin1 and the new results have been included in the revised Figure 4B. When added before the addition of *C. albicans*, laminarin impaired phagocytosis, consistent with the involvement of Dectin1 in the initial stages of the interaction, which was validated by comparing RAW cells (that express little Dectin1) with RAW-Dectin1 cells, which are stably transfected to express higher levels of Dectin1 (new Figure 4A). However, when added after *C. albicans* adherence, laminarin had no effect on actin cuff formation (Figure 4B). As discussed in the manuscript, we believe that at this later stage the integrins, specifically CR3, have been activated and become the main drivers of the association and of the formation of the actin cuff.

7) Since the mannose receptor is likely to engage with hypha mannan, (which is located in the outer cell wall) it would be interesting to know if this is located at this site.

The possible involvement of mannan receptors was tested in several ways. First, as mentioned above, we used mutant strains that are mannan-deficient. Secondly, we tested the effects soluble mannans on the internalization of *C. albicans* and the formation of actin cuffs and have included these data in the new Figure 4B. In accordance with the mutant data, we found soluble mannan to be without effect. Lastly, we immunostained for mannose receptors using antibody MR5D3 (Bio-Rad; catalogue # MCA2235GA) and found no evidence of accumulation at the cuff. These new results are described in subsection “Role of receptor cooperativity in actin cuff formation” of the revised text.

8) I'm not sure that the TEM shown in Figure 1C can be seen as evidence of showing the actin cuff unless the actin is perhaps stained using colloidal gold.

We agree that the electron micrographs do not provide direct evidence that actin is accumulated in the region of cuff, a case much better made by the phalloidin staining that is profusely illustrated throughout the paper. We clearly overstated the case in the original version and have toned down the interpretation of the images in the revised text. Instead, we refer to the similar exclusion area seen by Strijbis et al., (2013), who used tannic acid negative staining to better visualize actin (see subsection “Phagocytosis of *C. albicans* hyphae” of revised text).

9) My88-/- cells would be useful in assessing the possible involvement of TLRs.

The involvement of TLRs/MyD88 in the recognition of and control of *C. albicans* by macrophages is well documented. Indeed, Marr et al., (2003) showed that MyD88^−/−^ cells have a pronounced phagocytic defect. While reflecting the importance of the cooperation between MyD88/TLR and Dectin1, this phagocytic defect would complicate performing the suggested experiments using MyD88^−/−^ cells, as well as their interpretation, since it would be difficult to separate the contribution of MyD88 to actin cuff formation from that to Dectin1-mediated phagocytosis. While we have included additional experiments to dissect the contributions of Dectin1 and CR3, we feel that additional studies of MyD88 and TLRs would be outside the scope of this manuscript.

10) The actin cuffs appear to be elongated in some images and much shorter or almost like spots in others. Do they retain their shape over time or contract to a spot?

It is our observation that the actin cuffs can vary somewhat in length and continuity from cell to cell, sometimes fragmented into two or more smaller cuffs. However, once established in any single cell, the actin cuff does not change shape or contract significantly after formation for up to 90 min (the duration of our experiments), despite the fact that actin treadmilling is ongoing at the cuff. We refer the reviewer to subsection “Phagocytosis of *C. albicans* hyphae” of the revised text, where these observations are described.

Reviewer #2:The authors claim that the frustrated phagosomes are microbicidal. Yet, there is no data on fungal survival to support this conclusion. Are the macrophage-trapped hyphae growing longer or dying as time passes?

This point was also raised, equally aptly, by reviewer 3. To address this issue, we conducted new experiments to assess the antimicrobial effects of the actin cuff on *C. albicans* hyphae. We initially measured the viability of the hyphae that had been partially internalized using propidium iodide. As is now described in the Results section, we did not see any change in fungal viability in the partially internalized *C. albicans* hypha (nor in fully internalized hyphae, for that matter; data not shown) for at least 60 min, the normal duration of our experiments. We reasoned that the antimicrobial effectors may not suffice to kill the fungus yet may curtail hyphal growth. The remarkable extension rate of *C. albicans* hyphae, which has been well-documented (18.8 µm/hr = 0.31 µm/min, Gow, 1985), enables accurate quantitation during the course of our experiments. We therefore, compared the rate of growth of partially internalized hyphae with an actin cuff with that of hyphae that were fully internalized or that were not engaged by macrophages (i.e. remained extracellular) throughout. These results are now summarized in the new Figure 10F and described in subsection “Functional properties of the frustrated phagosome” of the revised text. Briefly, we found that sequestration of a part of the hyphae within the frustrated phagosomes delimited by the cuff reduced the rate of growth considerably, to the same extent as seen for hyphae fully internalized within a sealed phagosome. Importantly, blocking of CD11b with the M1/70 antibody abolished this microbiostatic effect (Figure 10—figure supplement 2). In light of this newly characterized effect on hyphal growth, we have changed the title from “microbicidal” to “microbiostatic”.

The contribution of Candida hyphae to the formation of the frustrated phagosomes and actin cuffs must be assessed. Changes in Candida cell wall composition have been previously reported to affect macrophage actin dynamics (Bain et al., 2014). However, the authors are not considering or discarding a role for the yeast in the proposed model.

As described above when addressing comment #4 of reviewer 1, we performed a new series of experiments to define the cell wall components required for actin cuff formation, using a curated set of *C. albicans* GRACE strain mutants obtained from Merck and from Genome Canada, which have mutations affecting biosynthetic pathways for chitin, mannans or β(1,6)glucans (see O’Meara et al., 2015). We also conducted pharmacological experiments targeting the formation of the β(1,3)-glucan bond, using caspofungin. In a nutshell, we found that β(1,3)glucan is required for actin cuff formation, while chitin, mannans and β(1,6)-glucans were dispensable. Please see the new Figure 4 (Figure 4C, D, E and F, Figure 4—figure supplement 1) and the respective text in subsection “Fungal cell wall components that contribute to actin cuff formation” of the revised manuscript for details.

For the imaging data, even though the regions of interest are nicely indicated with brackets, since different markers don't always localize to same regions along the frustrated phagosome, having additional panels with channels merged would make it easier to visualize what the authors are illustrating.

As suggested, to ease interpretation of the images, we have added ~ 2x merged insets of the actin cuff region, where the colocalization of various markers is indicated as yellow. See revised Figure 2, Figure 3, Figure 5, Figure 6 and Figure legends. In addition, we now provide quantitation of specific marker enrichment/exclusion in the actin cuff, calculated as the ratio of fluorescence in the cuff vs. the phagocytic cup (Results section, and Materials and methods section of the revised manuscript). Finally, we have added new 3D reconstructions to aid the viewer in visualizing the shape of the cuff and its relationship to the extracellular portions of the hyphae and to the phagosomal marker LAMP1 (see Figure 6G and H). Rotational views and progressive deconstruction of the images are provided in accompanying Video 2.

In subsection “Signals driving actin cuff formation”, referring to Figure 2AB, the authors indicate, a lack of Dectin-1 at the actin cuffs. However Emerald-Dectin-1 can be seen in the cuff in Figure 2B, which is in good agreement with Strijbis et al., (2013) that also showed that Dectin-1 is enriched in the cuff region of the phagocytic cup, as well as in areas of enrichment outside the cuff region. To support their claim, the authors must present quantitative data on Dectin-1 distribution along the frustrated phagosome.

As requested by the reviewer, we have quantitated the ratio of Dectin1-HA and Emerald-Dectin1 in the cuff compared to the cup; this quantitation shows that the receptor is indeed partially excluded from the cuff (see Results section). Moreover, we now provide insets to Figure 2A and B showing the localization of Dectin1 and merged images showing also the actin cuff. Strijbis et al., (2013) did assess Dectin1 localization and found that “Dectin1 was enriched in the cuff region of the phagocytic cuff….but also showed areas of enrichment outside the cuff region”. We believe that this is consistent with our observations that Dectin1 is present throughout the phagocytic cup but is not enriched in the area of the actin cuff, relative to the rest of the frustrated phagocytic cup. Additionally, please see point 5 below for further discussion of the localization of Dectin1.

Also, it is not indicated, but the imaging shown in Figure 2, and other figures, seems to be a single confocal z-plane. It may also be appropriate to include extended focus projections, considering the size of the hyphae, which could be included as supplementary files.Clarifying the distribution of Dectin1 in the frustrated phagosomes, is critical because it affects the interpretation of data in Figure 2D and other results in the manuscript.

We agree that clarifying the role of Dectin1 is important. To this end, we have added to Figure 2A and B insets showing the merged images and dotted outlines of the cells and have included the quantitation of Dectin1 density described above (see Results section). Moreover, as described under comment #3, we now include 3-dimensional reconstructions and videos of the cuff and its relationship to other components. We also attach Author response image 2, the extended focus projections of the confocal images illustrated as Figures 2A and B, for the reviewer’s perusal. We do not feel that inclusion of such extended focus images would be more informative but would certainly include them as part of Figure 2—figure supplement 1 if the reviewer feels this is useful.

**Author response image 2. respfig2:** 

Figure 2D showing PAK-PBD and tyrosine phosphorylation in the actin cuffs are likely expected results since Strijbis et al., 2013, have shown that the GEF for CDC42 and RAC1, VAV1, as well as the tyrosine kinases Syk and BTK, are all recruited to cuffs. Quantitative data for the recruitment of these molecules to the frustrated phagosomes and data proving that their activities are involved in the formation or sustaining the cuffs will certainly help the proposed model.

As suggested by the reviewer, we have quantitated the enrichment of PAK(PBD) and phosphotyrosine in the actin cuff region (see Results section and Materials and methods section). Additionally, we present a new Figure 5 where the contribution of distinct tyrosine kinases to actin cuff formation was assessed in the context of the frustrated phagosome anchored by CR3.

This extends and complements the work of Strijbis et al., (2013), who first described the role of Syk and BTK in the context of Dectin1-mediated phagocytosis. To this end, we used immunostaining to assess the activation of several candidate tyrosine kinases. Specifically, we have detected activation (phosphorylation) of Syk and, interestingly, PYK2/FAK kinases at the actin cuff (see Figure 5, and subsection “Signals driving actin cuff formation” of the text for quantitation). To our knowledge, PYK2 and FAK were previously not known to participate in the host response to *C. albicans*. These observations prompted us to assess the functional requirement of these kinases for cuff formation and maintenance. As illustrated in the new Figure 5E, inhibition of PYK2/FAK obliterated the formation of actin cuffs, as did inhibition of Syk. It is noteworthy that, by contrast, Src-family kinase inhibitors were without effect.

To extend the findings of Strijbis et al., (2013) regarding actin polymerization, we performed new experiments assessing the contribution of Arp2/3- vs. formin-dependent actin assembly to actin cuff formation, since both can be driven by active Rac1 and Cdc42 GTPases. As shown in the new Figure 5G we found that formins are the primary driver of actin cuff formation. These data provide the basis of a more complete model of cuff formation and are consistent with known pathways driving actin polymerization after engagement of CR3 (see Results section and Discussion section).

Figure 3D shows that on average, for a given field, 6 hyphae are observed in frustrated phagosomes with actin cuffs and this is consistent throughout the manuscript. However, it is not clear how many individual events were analyzed. How frequently are these frustrated phagosomes observed? Do all hyphae engaged by macrophages have these structures associated with them?

In accordance with the reviewer’s request we have included quantitation of the frequency of actin cuff occurrences in the text (see Results section). As mentioned in the revised text, under the conditions used, partially internalized hyphae occur at a frequency of 68.5%, and of those 96.3% have actin cuffs. In addition, we now provide the average number of *C. albicans* hyphae visualized per field, to accompany the graphical data (see Figure legends), and have amended all graphical figures to show the number of *C. albicans* with an actin cuff, along with the number of fully internalized hyphae, to better reflect the total number of *C. albicans* events analyzed per field (see Figure 2, Figure 3, Figure 4 and Figure 5).

Figure 4. For accumulation of receptors provide quantitative data. According to the authors, MAC1 activation signals for the formation of the actin cuffs. However, in Figure 4A, CD11b did not localize to the thicker section of the cuffs. Please clarify and indicate the frequency of this phenotype.

To provide further clarity as to the localization of CD11b and CD18, we now provide merged ~2x insets showing the colocalization (in yellow) of CD11b and CD18 with the actin cuff (see Figure 3A and B), and we have quantitated the enrichment of CD11b and CD18 in the actin cuff region (see Results section and Materials and methods section for calculations). Note that, in other integrin-induced structures such as stress fibres and podosomes, the regions of greatest actin accumulation are located at some distance from the active integrins themselves. It is also possible that access to the epitope (which is exofacial in both instances) may be restricted by the close apposition of the host and pathogen surfaces, particularly in those areas where the actin cuff may constrict the neck.

Figure 4G shows a shorter cup in the presence of CD11b blocking antibodies. The treatment could cause an inhibition of phagocytosis. Authors must demonstrate that phagocytosis is not affected by the treatment.

As the reviewer has requested, we have amended the former Figure 4H (now Figure 3H) to include the number of fully internalized *C. albicans*, in addition to those that are partially internalized and therefore display actin cuffs. As the new figure indicates, the number of fully internalized hyphae, a process largely mediated by Dectin1, was not significantly affected. The decrease in the number of actin cuffs seen upon inhibition of CD11b was accompanied by a reciprocal increase in the number of partially internalized hyphae that did not display actin cuffs, again implying that engagement of the hyphae and initiation of phagocytosis did not require CR3. To simplify the graph in Figure 3H, the number partially internalized without actin cuffs has not been included but is referred to in the text (see Results section).

Figure 4G/H In cells treated with the M1/70 antibody, actin is still present in the frustrated phagosome. Is this a cuff? In fact, Concavalin A labeling is limited to the phalloidin boundary. This suggests the presence of a cuff/ barrier, even after CD11b treatment.

The original Figure 4G (now Figure 3G) is representative of the observed disruption of the actin cuff around partially internalized *C. albicans* hyphae. As should be apparent, the amount of actin is greatly reduced and, more importantly, the entire cup is lined by actin, unlike the discrete distribution at the cuff, which is now clearly illustrated in the 3-dimensional images shown in Figures 1E-H and Video 1. It is important to note that preferential labeling of the extracellular portions of the hyphae by Concanavalin A (ConcA) is not predicated on the barrier function of the integrin/actin cuff. Indeed, the Stokes radius of ConcA, reported as 3.0 nm (Sawyer et al., 1975; Ahmad et al., 2007) is similar that of 10 kDa dextran (2.3 nm), and considerably smaller than that of the 100 kDa dextran (>6 nm), which was shown not to traverse the junction. It is the result of the comparatively brief (20 min) incubation at reduced (room) temperature with the lectin, which favors readily detectable binding to exposed carbohydrates, while minimally staining those inside the cup, where diffusion is limited by the narrow spacing between the membrane and hyphal wall. Moreover, ConcA staining is done after fixation with paraformaldehyde. Paraformaldehyde crosslinking likely restricts further the entry of Concanavalin A to the regions trapped in the cup. Nevertheless, to address the effectiveness of the block exerted by antibody M1/70, we took advantage of the observation that formation of the cuff impaired the growth of the hyphae. We performed new experiments where the rate of growth was compared in hyphae that were partially internalized by cells in the presence and absence of antibody M1/70. The new data, which are presented in Figure 10—figure supplement 2, show that blocking the engagement of CR3 eliminated the microbiostatic effect of the macrophages, i.e. the hyphae grew at normal rates in M1/70-treated cells. See response to comment #2 by reviewer 3 for further description of these data.

Is dectin1 distribution along the frustrated phagosome affected after CD11b is blocked? Blocking CD11b in dectin1 knockout cells will help address whether dectin1 contributes to this barrier.

As requested by this reviewer, as well as by reviewer 1, we performed additional experiments to assess the role of Dectin1 in actin cuff formation. We took advantage of the earlier observation that Dectin1 is essential for *C. albicans* internalization(Taylor et al., 2007, Marakalala et al., 2013 –these references have now been cited in subsection “Role of receptor cooperativity in actin cuff formation”). Based on this knowledge we designed experiments comparing the Dectin1-deficient (parental) RAW 264.7 cell line with the RAW-Dectin1 stable transfectants. Importantly, while cuff formation was minimal in the Dectin1 deficient cells, we could bypass the need for the glucan receptor using complement opsonization (see new Figure 4A, discussed in subsection “Role of receptor cooperativity in actin cuff formation” of the revised text). We interpret these results to mean that Dectin1 is required for initial engagement and initiation of phagocytosis, a role that can be alternatively fulfilled by the traditional integrin-binding domain of CR3 yet is not essential for cuff formation. Because in the absence of complement phagocytosis will not occur in Dectin1 knockout cells, the blocking experiment suggested by the reviewer cannot be performed. Instead, we timed the inhibition of Dectin1 using laminarin, as recommended by reviewer 1. When added before phagocytosis, laminarin greatly inhibited cuff formation, as expected, but it was ineffective when added after the integrin was already engaged (Figure 4B). Note that size of the laminarin used (≈500 Da) was sufficiently small for it to penetrate the junction between the macrophage and the hyphae, based on the data of Figure 10.

Figure 5, subsection “Phospholipid segregation between the plasma membrane and the cuff-delimited phagosomal cup” states "Astonishingly, while PtdIns(4,5)P2 was present as expected in the surface membrane facing the extracellular milieu, it was undetectable in the invaginated section that constituted the frustrated phagosome (Figure 5A)". This and other characteristics of the cuff and the frustrated phagosomes reported in Figure 5 and Figure 8 are similar to those reported by Prashar et al., (2013). This must be acknowledged in the manuscript. It must be acknowledged that PtdIns(3,4,5)P3 distribution in the frustrated phagosome has been previously described by Strijbis et al.

At the reviewer’s request, we have cited these references in the appropriate context in several places in the text. Please see Results section and Discussion section).

In Figure 5D, how do the authors explain the presence of LC3 in the actin cuff?

LC3 has been reported to insert into the phagocytic cup during LC3-assisted phagocytosis (see Martinez et al., 2015 reference in Results section), a maturation pathway that has been implicated as important for the control of fungus, including *C. albicans* (Kanayama et al., 2016, Tam et al., 2016, Sprenkeler et al., 2016 –references now added to the text, see Results section). We chose to analyze its distribution because it is lipid-anchored yet can be covalently labeled by a fluorescent protein. The precise mechanism that directs LC3 to phagosomes is not known and, most importantly, we do not know whether these determinants are deployed in the area of the membrane where the integrins generate the cuff, which would readily explain its presence there. Most importantly, as discussed in more detail below, the diffusion barrier may slow down the movement of lipids, lipid-anchored proteins and proteins markedly, without necessarily preventing their entry into the cuff altogether. In this instance, the much faster diffusion of these molecules after they exit the cuff would make them undetectable outside the cup.

Subsection “The actin cuff forms a diffusional barrier to the movement of proteins and lipids” "The sharp boundary between the PtdIns(4,5)P2-rich surface membrane and the tubular membrane endowed with PtdIns(3,4,5)P3 and PtdIns(Bauer et al., 2001)P coincided with the location of the actin cuff, suggesting that the latter may function as a diffusion barrier".Subsection “The actin cuff forms a diffusional barrier to the movement of proteins and lipids” Similarly, both wild-type Rab7 (Figure 5E) and constitutively-active Rab7 (not illustrated) are confined to the frustrated phagosomal tube, as was LAMP1 (Figure 5F)[…]we considered it more likely that restricted diffusion accounted for the observations"Differently to what the authors describe in the paragraph from above, the phagosomal markers are penetrating the barrier as in the case of Figure 5B and E and surpassing the cuffs in Figure 5C, E and F. As indicated before for other results the authors must present quantitative data on the distribution of the markers and extended focus projections to support the proposed model.

This is an important issue that indeed required clarification. First, to more clearly illustrate the relative disposition of the markers vis-a-vis the actin cuff, we have added ~ 2x merged insets of the cuff region where the colocalization of various markers with actin is indicated in yellow (see revised Figure 6). As we described in the original version, the boundary of the lipids coincided with the location of the actin band, although in the case of PtdIns(4,5)P_2_ the inositide is restricted to the plasmalemma, while PtdIns(3,4,5)P_3_ is both in the cup and throughout the cuff. It is quite likely that PI3-kinase is active in the region where the integrins and active Syk and PYK2/FAK accumulate, so that PtdIns(3,4,5)P_3_ could be generated within the cuff itself. As stated above, in the case of the lipids in particular, it is likely that the barrier reduces the diffusion rate of the lipids considerably, without necessarily preventing their entrance into the region of the cuff altogether. The limited amounts of PtdIns(4,5)P_2_ entering the cuff would be converted to PtdIns(3,4,5)P_3_ and/or hydrolyzed by PLC. This, we believe, accounts for the differential appearance of the two inositides.

Also, as requested, we have now provided quantitation of marker enrichment/exclusion in the actin cuff, calculated as the ratio of fluorescence in the cuff vs. phagocytic cup (see Results section and Materials and methods section). Based on these measurements we have modified the wording of that section as follows:

“Similarly, both wild-type Rab7 (Figure 6E) and constitutively-active Rab7 (not illustrated) are confined to the frustrated phagosomal tube and partially excluded from the actin cuff (Rab7 ratio cuff: cup 0.68 ± 0.05; n=30, p<0.0001), as was LAMP1 (ratio cuff: cup 0.59 ± 0.03; n=30, p<0.0001; Figure 6F)”.

The partial exclusion of the marker proteins is consistent with molecular crowding caused by integrin accumulation at the cuff, which we propose causes reduced penetration and diffusion within the area of the cuff.

Figure 7 strongly supports authors' claim However, if latranculin A treatment allowed Lamp-1 and PtdIns(4,5)P2 to migrate to previously excluded zones, why concanavalin A is not following and labels the full extension of the hyphae? That will be the expected if the diffusion barriers in the cuffs are dismantled. Quantitative data is missing, and full cell imaging would be preferable to better illustrate the authors' claim.

As we described above in response to comment #9 by reviewer 2, preferential labeling of the extracellular portions of the hyphae by Concanavalin A (ConcA) is not predicated on the barrier function of the integrin/actin cuff. It is the result of the comparatively brief (20 min) incubation at reduced (room) temperature with the lectin, which favors readily detectable binding to exposed carbohydrates, while minimally staining those inside the cup, where diffusion is limited by the narrow spacing between the membrane and hyphal wall. Moreover, ConcA staining is done after fixation with paraformaldehyde. Paraformaldehyde crosslinking likely restricts further the entry of Concanavalin A to the regions trapped in the cup.

As requested, we have included quantitative data showing the effects of latrunculin A on the distribution of PLCδ-PH-GFP and Lamp1-GFP (new Figures 9C and F), supporting our observations depicted in the representative figures (Figures 9A, B, D, E).

Figure 8 The characteristics of the frustrated phagosome are remarkably similar to long cups described by Prasher et al., 2013 and Strijbis et al., 2003. This must be cited in the text.

In accordance with the reviewer’s request, we have cites these references in several places in the revised manuscript (see Results section and Discussion section).

The accumulation of NBT in Figure 8E is puzzling since small molecules can diffuse out across the actin cuff barrier. The authors must address the possibility that the NBT is accumulating inside the hyphae. NBT can accumulate in fungi Camile et al., 2008. Thus, authors must prove a link between ROS and microbicidal conditions in the long open phagosomes.

We were aware that the *C. albicans* can metabolize NBT, producing formazan. Bearing this in mind we performed the experiments reported in the original Figure 8 (now Figure 10 of the revised manuscript) using killed bacteria. Note that we tested both heat-killed and paraformaldehyde-killed *C. albicans* hyphae to ensure that the production of formazan was not dependent on the structure of molecular patterns that might have been affected by the method used. While these details were described in the Materials and methods section of the original version, we have now mentioned them also in the Results section to avoid confusion (see Results section and also Materials and methods section)

Reviewer #3:The manuscript by Maxson et al., describes and analyzes partially sealed phagocytic cups in macrophages which form around Candida albicans hyphae. It demonstrates that an actin-rich cuff at the distal margin of the cup supports a lateral segregation between the components of the plasma membrane and the contiguous inner leaflet of the phagocytic cup. This lateral segregation excludes the PI(4,5)P2 from the cup and confines the PI(Bauer et al., 2001)P and PIP3 to the cup. The actin cuff contains the integrin CR3 and associated integrin signaling molecules. The barrier effectively limits the movement of inner leaflet probe molecules into or out of the cup, without significantly diminishing diffusion within the plasma membrane or the cup. The actin cuff can be disrupted by the actin depolymerizing drug latrunculin A. The barrier limits escape by diffusion of large macromolecules delivered into the lumen of the cup, but not the diffusive loss of smaller molecules, including dextrans and protons (pH). Nonetheless, reactive oxygen species can be delivered into the unclosed cups. The experimental work is carefully done and the morphological evidence in support of the claims is beautiful. However, much if not most of the conclusions reported here were first described in a large study by Prashar et al., (2013), using an analogous experimental model. Examining the interactions between macrophages and filamentous bacteria, that study demonstrated (Astarie-Dequeker et al., 1999) the actin cuff (called a "jacket") that segregates two domains of contiguous membrane (plasma membrane and phagocytic cup), (Bain et al., 2014) the effect of actin depolymerization on the maintenance of that segregation, (Bauer et al., 2001) the nature of the diffusion barrier with respect to extracellular probes and (Belin et al., 2014) the retention of diffusible molecules in the cup lumen (different sizes of dextrans, pH, hydrolytic enzymes).

In accordance with the reviewer, we have cited this reference repeatedly throughout the text to more accurately lay the ground for our studies (see Results section and Discussion section). It is important to point out that our study extends and complements the Prashar et al., study in several ways: (Astarie-Dequeker et al., 1999) we identify the receptors involved in generating the actin cuff; (Bain et al., 2014) using a variety of mutants and pharmacological agents, we now identify the ligand recognized by the lectin domain of the integrin, namely the β(1,3)-glucan bond (see new Figure 4); (Bauer et al., 2001) we describe experiments indicating that molecular crowding is responsible for the diffusion barrier (see new Figure 8); (Belin et al., 2014) we describe the activation and functional role of PYK2/FAK in the phagocytosis of *C. albicans* hyphae (see new Figure 5); (Ben-Ami et al., 2011) we document actin turnover (treadmilling) at the cuff, despite the maintenance of the net amount of actin over extended periods; (Benard et al., 1999) we assign a primary role to formins, as opposed to Arp2/3 in the process (see new Figure 5G); (Bergmeier et al., 2007) we demonstrate that molecules trapped in the phagocytic cup remain mobile; (8) we demonstrate that generation of the cuff has a microbiostatic effect, limiting the growth of the hyphae and (9) we highlight the differential behaviour of lipid/lipid-anchored molecules located in the inner and outer leaflets of the membrane.

Moreover, unlike the present manuscript which shows that ROS can be generated in the unclosed cups, the previous paper measured the effect of partial phagocytosis on microbicidal activities against filamentous bacteria (albeit somewhat incompletely). The present manuscript but does not measure microbicidal activity against C. albicans.

This point was also raised, equally aptly, by reviewer 2. To address this issue, we conducted new experiments to assess the antimicrobial effects of the actin cuff on *C. albicans* hyphae. We initially measured the viability of the hyphae that had been partially internalized using propidium iodide. As is now described in the Results section, we did not see any change in fungal viability in the partially internalized *C. albicans* hypha (nor in fully internalized hyphae, for that matter; data not shown) for at least 60 min, the normal duration of our experiments. We reasoned that the antimicrobial effectors may not suffice to kill the fungus yet may curtail hyphal growth. The remarkable extension rate of *C. albicans* hyphae, which has been well-documented rate (18.8 µm/hr = 0.31 µm/min, Gow, 1985), enables accurate quantitation during the course of our experiments. We therefore, compared the rate of growth of partially internalized hyphae with an actin cuff with that of hyphae that were fully internalized or that were not engaged by macrophages (i.e. remained extracellular) throughout. These results are now summarized in the new Figure 10F and described in subsection “Functional properties of the frustrated phagosome” of the revised text. Briefly, we found that sequestration of a part of the hyphae within the frustrated phagosomes delimited by the cuff reduced the rate of growth considerably, to the same extent as seen for hyphae fully internalized within a sealed phagosome. Importantly, blocking of CD11b with the M1/70 antibody abolished this microbiostatic effect (Figure 10—figure supplement 2, and Results section). In light of this newly characterized effect on hyphal growth, we have changed the title from “microbicidal” to “microbiostatic”.

Assuming this manuscript can be rewritten to acknowledge appropriately the demonstrated precedents and concepts of that earlier work, the present study does add some interesting new data which supports a role for integrin in the establishment or maintenance of the actin cuff.

As mentioned above, we have rewritten the text to better assign credit to earlier reports, while highlighting our novel findings. The most salient novel contributions are itemized in the response to comment #1 by this reviewer. We feel that the new data and insights presented represent a significant advance to the study of phagocytosis and host-fungus interactions.

The role of integrin in the maintenance of the barrier remains indirect, however. Perhaps the actin ring is the main ingredient of the barrier, and anything which organizes actin into a ring can support the formation of diffusion barriers such as described here, and elsewhere for analogous structures Golebiewska et al., 2011; Welliver et al., 2011. To better define the nature of the diffusion barrier, the diffusion measurements (e.g. Figure 6) or probe localization studies (e.g. Figure 5) described in the manuscript should be extended to analyze the roles of actin (latrunculin A) and integrin (M1/70) to barrier maintenance. This could provide a mechanistic underpinning to the diffusion measurements.

The reviewer raises an important and valid point, which we have tried to address by performing a series of new experiments. As intimated (but not demonstrated) in the original version, we hypothesized that the diffusional barrier is the consequence of molecular crowding, generated by clustering of CR3 and associated molecules to a high density. Dense clustering was envisaged to occur upon exposure to a profusion of β(1,3)-glucan bonds on the hyphae. To test this hypothesis, we induced crowding of CR3 by crosslinking the integrins with antibodies (in the absence of *C. albicans* hyphae), adapting the traditional technique used previously to patch/cap surface receptors (see new Figure 8B for diagrammatic illustration). Having successfully clustered CR3, we analyzed whether this sufficed to exclude other proteins.

To this end, we transfected the cells with a fluorescent transmembrane protein (CD2-CD45GFP) and quantified the relative distribution of the two proteins before and after cross-linking CR3. As shown in the new Figure 8, patches formed by clustering CR3 excluded CD2-CD45-GFP, implying that a diffusion barrier had been generated by increasing the local density (crowding) the integrin. Importantly, neither formation of the patches, nor the exclusion of CD2CD45-GFP required actin, since latrunculin was without effect. These observations imply that clustering of the transmembrane protein CR3, as opposed to its effects on actin recruitment, are the primary determinant of the diffusion barrier. This conclusion is in good agreement with the realization that transmembrane “pickets”, rather than the underlying actin “fence”, are the primary obstacles to the diffusion of membrane-associated proteins and lipids (Freeman, et al., 2018, in press).

It is important to bear in mind that, while actin itself may not be required to exclude mobile proteins and lipids form areas of CR3 crowding, it is nevertheless required to establish and maintain the clusters, ostensibly by stabilizing the active form of CR3 via talin and vinculin, as described for other integrins. This explains why the diffusional barrier generated by interaction with hyphae disassembles when the cells are treated with latrunculin A (Figure 9). The new experiments are now described in subsection “Examining the role of CR3 and actin in the maintenance of diffusional barriers” of the revised paper.

[Editors’ note: what now follows is the decision letter after the authors submitted for further consideration.]

1) In the abstract it is stated “[…]in response to non-canonical activation of integrins by fungal β(1,3)-glucans." We suggest using the singular "glycan". There is more than one form of fungal β(1,3)- glucan, but this is not relevant to this article.

We have changed the text as requested; please see the Abstract of the revised version.

2) B-glucans are involved in the activation of complement and the deposition of C3b fragments on the surface of C. albicans. (Boxx et al., 2010., Kozel et al., 1987, Vetvicka et al., 1996). What could be the contribution of C3b opsonization on the formation of the actin cuffs and the properties of the membrane and luminal diffusion barriers?

In the absence of serum opsonization, the observed phenomena are dependent on engagement of Dectin1, which secondarily enables the activation of integrins that bind to glucan. However, when Dectin1 expression is low, complement deposition suffices to engage the integrins and directly activate this process. We have observed that the actin cuffs created via C3b opsonization occur with similar frequency and appearance, therefore we assume that in our experiments and in the host, C3b deposition has a parallel role to Dectin1. We have included text in the Discussion section to elaborate on this point.

3) Previous reports on the phagocytosis of C. albicans showed the formation of PIPs in phagosomes containing this yeast (Heinsbroek, 2009). This was attributed to pathogenic mechanisms. On the other hand, and coinciding with the authors' interpretation, Naufer et al., 2018 recently reported that PI(Bauer et al., 2001)P co-exists with phagolysosomal markers in the phagocytic cup of heat-killed filamentous bacteria. Maybe the authors could mention this in their Discussion section.

As reported by Heinsbroek et al., 2009, we have also seen secondary waves of actin polymerization as a result of reacquisition of PI(4,5)P_2_ and PI(3,4)P_2_/PI(3,4,5)P_3_ on the phagosomes of serum opsonized sheep red blood cells (Bohdanowicz et al., 2010), internalized via CR3. We have added additional text and references to both papers in subsection “Phospholipid segregation between the plasma membrane and the cuff-delimited phagosomal cup” of the revised manuscript. We have also mentioned the work of Naufer et al., 2018 to indicate that live *C. albicans* is not required for actin cuff formation (as we see in Figure 10E). This new paper is also now referenced in subsection “Phospholipid segregation between the plasma membrane and the cuff-delimited phagosomal cup”. However, we feel that extensive discussion of the similarities/differences between these findings and our work is outside the scope of this manuscript.

4) Please clarify: In Figure 4, human serum was used to complement opsonize hyphae. How did the authors account for the presence of antibodies against C. albincans in the serum? Antibodies could be expected as this yeast is part of the human microbiome

The reviewers raise a valid point; in the text and figures we have purposefully used the general term “serum opsonization” and not complement opsonization, because additional opsonization by antibodies to *C. albicans* could indeed occur. This is now explicitly acknowledged in the Discussion sectionof the main text, and also in the Materials and methods section. Although not mentioned in the paper, we believe that immunoglobulins contribute comparatively little to the opsonization because Fc receptor-dependent phagocytosis is dependent on SFKs and Arp2/3, while we find the formation of actin cuffs to be insensitive to the respective inhibitors PP2 and CK666 (Figures 5E and 5G).

5) Caspofungin treatment could prevent the deposition of complement in hyphae, and its recognition by antibodies and Dectin-1. This could explain the results in Figure 4—figure supplement 1.

We thank the reviewers for their astute observation. We have included discussion of this caveat in the Discussion section. It is possible that complement binds less to the caspofungin-treated hyphae, as reports of Boxx et al., 2010 and Kozel et al., 1987 indicate that a portion of the complement binding occurs on fungal glucans. Nevertheless, we regard this as unlikely because the fungal cell wall is rich in other polysaccharides and proteins that can serve to attach complement. In this regard, it is noteworthy that caspofungin-treated *C. albicans* bind concanavalin A normally (Figure 4E), indicating that mannans are still present.

6) The authors showed that ROS production in the phagocytic cup retarded the elongation of the hyphae. This was alleviated by blocking CR3 with monoclonal antibodies to impede the formation of cuffs, a procedure that favours a ROS leaching. Since ROS are generally considered microbicidal, perhaps, ROS failing to reach lethal concentrations in the open cup could be the cause of the effect reported by the authors. Therefore, assuming a microbiostatic mechanism is probably incorrect.

As pointed out be the reviewers, when CR3 is blocked ROS may exit the open cup more readily, reducing their efficiency. However, leakage of phagosomal contents or ROS does not explain the failure of the frustrated phagosomes to kill the fungus, because *C. albicans* yeast and hyphae survive also within fully sealed phagosomes. Thus, in the case of *C. albicans*, ROS is not strictly microbicidal. This is now discussed in the Discussion section of the revised text.

Additionally, after reviewing the changes requested for submission of the revised manuscript, we now ensure that the requested file formats are provided, the title has been reconciled, the requested cell line information is now included in the Materials and methods section, and the source data for graphs in the main figures is provided. Please note, that in lieu of a Key Resources Table and RRIDs, we have comprehensively provided all essential information (suppliers and catalogue numbers for all cell lines, antibodies, and plasmids and reagents) in the Materials and methods section.